

# Impact of different estimations of the background-error covariance matrix on climate reconstructions based on data assimilation

Veronika Valler[1,2], Jörg Franke[1,2], and Stefan Brönnimann[1,2]

[1]Institute of Geography, University of Bern, Bern, Switzerland
[2]Oeschger Centre for Climate Change Research, University of Bern, Bern, Switzerland

**Correspondence:** Veronika Valler (veronika.valler@giub.unibe.ch)

**Abstract.** Data assimilation has been adapted in paleoclimatology to reconstruct past climate states. A key component of the assimilation system is the background-error covariance matrix, which controls how the information from observations spreads into the model space. In ensemble-based approaches, the background-error covariance matrix can be estimated from the ensemble. Due to the usually limited ensemble size, the background-error covariance matrix is subject to the so-called sampling error. We test different methods to reduce the effect of sampling error in a published paleo data assimilation setup. For this purpose, we conduct a set of experiments, where we assimilate early instrumental data and proxy records stored in trees, to investigate the effect of 1) the applied localization function and localization length scale; 2) multiplicative and additive inflation techniques; 3) temporal localization of monthly data, which applies if several time steps are estimated together in the same assimilation window. We find that the estimation of the background-error covariance matrix can be improved by additive inflation where the background-error covariance matrix is not only calculated from the sample covariance, but blended with a climatological covariance matrix. Implementing a temporal localization for monthly resolved data also led to a better reconstruction.

## 1 Introduction

Estimating the state of the atmosphere in the past is important to enhance our understanding of the natural climate variability, the underlying mechanisms of past climate changes and their impacts. To infer past climate states, two basic sources of information are available: observations, and numerical models. Climate models constrained with realistic, time-dependent boundary conditions provide fields that are consistent with the external forcings and the model physics. Observations can be instrumental meteorological measurements, which are mainly available from the mid 19th century. Prior to this time, information from proxies stored in natural archives (like trees, speleothems, marine sediments, ice cores) or documentary data can be exploited. Observations provide important local information, however their spatial and temporal coverage is sparse.

In recent years, a novel technique, the data assimilation (DA) approach, has been adapted for paleoclimatological research. DA creates a framework to combine information from different sources. If information from observations is optimally blended with climate model simulations, the result is the best estimate of the climatic state, given the observations, given the boundary conditions, and given the known climate physics. The field of paleo data assimilation (PDA) has undergone profound develop-





ments, and many DA techniques have been implemented to reconstruct past climate states, such as forcing singular vectors and pattern nudging (Widmann et al., 2010), selection of ensemble members (Goosse et al., 2006; Matsikaris et al., 2015), particle filters (e.g., Goosse et al., 2010), the variation approach (Gebhardt et al., 2008), the Kalman filter and its modifications (e.g., Bhend et al., 2012; Hakim et al., 2016; Franke et al., 2017; Steiger et al., 2018). However, there are still unresolved problems,

and thus, a need for improvements how to best combine observations with climate model simulations.

One popular DA method is the Kalman filter (KF; Kalman, 1960). The KF provides an estimate of the state that can be shown to be optimal with linear models and Gaussian distributions (Ghil and Malanotte-Rizzoli, 1991). In standard applications, the processes of the KF can be summarized in two main steps (Ide et al., 1997). In the update step, the background state and the uncertainty of the background state provided by the model simulation are adjusted by assimilating new observations. In the

forecast step, the updated state, called the analysis, and the uncertainty of the analysis are propagated forward in time. These processes are repeated when new observations become available. However, in PDA, the forecast step is usually neglected, that is the filter is used offline (e.g., Franke et al., 2017). Because the process is not cycled, the background state is obtained from a pre-computed model simulation. In some previous PDA studies, the background state is constructed once from the model simulation, and later, the same state is used in every assimilation window (Steiger et al., 2018, and references therein). In other

PDA studies, the background state is specific for the current assimilation window, that is, the state changes in each assimilation window according to the forcings (Bhend et al., 2012; Franke et al., 2017).

An essential component of the KF is the uncertainty of the background state. The true climate state is not known, therefore it has to be estimated. In ensemble-based approaches, an ensemble of the background state provides estimation of the truth, represented by the ensemble mean, and the perturbations from the mean are used to estimate the uncertainty, represented by

the background-error covariance matrix. Ensemble-based KFs are simplification of the KF, because the true state is usually sampled with a few tens to a few hundreds of ensemble members. The limited ensemble size leads to errors in the estimation of the background-error covariance matrix. This effect is known as the sampling error.

Two methods are commonly used to reduce the negative effect of sampling error: inflation (e.g., Anderson and Anderson, 1999), and localization (e.g., Hamill et al., 2001) of the background-error covariance matrix. A simple inflation technique is the

multiplicative inflation (Anderson and Anderson, 1999). Multiplicative inflation helps to maintain a more realistic distribution of the ensemble members by increasing the deviation of the members from the ensemble mean at each DA cycle (Anderson and Anderson, 1999), which is of minor importance in offline approaches. Covariance inflation, besides reducing the sampling error, also accounts for underestimated model error. In the additive inflation technique, the covariances are inflated by e.g., adding an additional error term to the background-error covariances (Houtekamer et al., 2005). Covariance localization removes

long-range spurious covariances in the background-error covariance matrix that occur by chance due to a limited sample size. Several localization techniques have been proposed, from a simple cut-off radius approach (Houtekamer and Mitchell, 1998) to more sophisticated ones (Houtekamer and Mitchell, 2001; Hamill et al., 2001). By applying covariance localization methods, the elements of the background-error covariance matrix are modified, and in the standard approach the covariances are forced to approach zero at a certain separation length from the location of the observation. This is achieved by multiplying



the background-error covariance matrix element-wise with a distance-dependent function. In practice, this function is often estimated by a Gaussian localization function, recommended by Gaspari and Cohn (1999).

In PDA studies, the time-dependent background-error covariance matrix is often replaced by a constant covariance matrix (e.g., Steiger et al., 2014). By using a constant background-error covariance matrix in the update step, the dependence on the climate state is lost. However, it is possible to estimate the covariance matrix from a much larger ensemble size, which reduces the sampling error. If the constant covariance matrix is built from a large enough sample size, representing different climate states, it can be successfully used in the assimilation process (Steiger et al., 2014).

Covariance inflation and localization techniques are used and under improvement in weather forecasting (e.g., Bowler et al., 2017), but have not been yet sufficiently explored for PDA. In this paper, we discuss three possibilities to improve the estimates of background error, relevant to our PDA method:

- using a two-dimensional multivariate Gaussian function as a horizontal localization function to test the hypothesis of longer correlation length scales in zonal than meridional direction.

- applying covariance inflation techniques. In the multiplicative inflation technique, a constant factor is used to inflate the deviations from the ensemble mean. In the other method, the background-error covariance matrix is calculated as the sum of the sample covariance matrix plus a climatological background matrix, where the climatological background is based on all ensemble members of multiple years. This larger sample size decreases the chances of spurious correlations.

- adding temporal localization to the background-error covariance matrix. Multiple time steps are combined in one assimilation window to efficiently assimilate seasonal paleodata. In case of monthly observations, covariances between the months have been used to update all six months (Franke et al., 2017).

This paper is structured as follows: An overview of our PDA approach, introducing the model, the observational network and the offline DA technique is given in Sect. 2. Section 3 describes the experimental framework. In Sect. 4 the results are presented and each experiment followed directly by a discussion. We summarize our experiments in Sect. 5.

## 2 Ensemble Kalman Fitting Framework

### 2.1 Model Simulation: CCC400

We start form an existing DA system, which is described in Bhend et al. (2012) and Franke et al. (2017). It uses a 30 member ensemble of atmospheric model simulations as background to reconstruct monthly climate states between 1600 and 2005. Simulations were performed with the ECHAM5.4 climate model (Roeckner et al., 2003) at a resolution of T63 with 31 levels in the vertical. The 30 ensemble members were forced with the same boundary conditions. For sea-surface temperatures (SSTs), which have a particularly large effect on the simulations, the reconstruction by Mann et al. (2009) was used. This is the only global gridded SST reconstruction that dates back till 1600. The SST reconstruction by design captures interdecadal variations (Mann et al., 2009), hence intra-annual variability dependent on a El Niño/Southern Oscillation reconstructions (Cook et al.,





2008) was added to the SST fields. Further boundary conditions include solar irradiance, land-surface parameters, volcanic activity, and greenhouse gas concentrations (for more details see Bhend et al., 2012; Franke et al., 2017). The 6-hourly output fields provided by the model were transformed to monthly means. To reduce the computational burden only every second grid points in the latitude and longitude were selected. We limit the analysis in this study to 2m-temperature, precipitation and
sea-level pressure.

## 2.2    Observational network

In this study, we use the same observational network of tree-ring proxies, documentary data and early instrumental measurements as described in Franke et al. (2017) (Fig. 1). The temporal resolution of the instrumental air temperature and sea-level pressure measurements, as well as the documentary temperature data, is monthly. The tree-ring proxy records have annual
resolution. Trees respond to a locally varying growing seasons. We consider temperature from May till August and precipitation from April till June to possibly affect tree-ring width data. The maximum latewood density proxies were considered to be affected by temperature over May till August. The observations were quality checked before the assimilation, and outliers which were more than 5 standard deviation away from the calculated 71-year running mean were discarded, for instrumental and proxy data. The documentary data were manually screened.

## 2.3    Assimilation method

In our paleoclimate reconstruction, we combine the CCC400 model simulation with the observations as described above by implementing a modified version of the ensemble square root filter (EnSRF; Whitaker and Hamill, 2002). This ensemble-based DA method is called ensemble Kalman fitting (EKF; Franke et al., 2017). In fact, the EKF is an offline version of the EnSRF. EKF is described in more detail in Bhend et al. (2012) and Franke et al. (2017). Here we shortly highlight the most important
aspects of the EKF. The update equation in the EnSRF scheme has two parts: updating the mean ($\overline{x}$), and for each member, the deviation from the mean ($x'$). They are calculated as

$$\overline{x}^a = \overline{x}^b + \mathbf{K}\left(\overline{y} - \mathbf{H}\overline{x}^b\right) \tag{1}$$

$$x'^a = x'^b + \tilde{\mathbf{K}}\left(y' - \mathbf{H}x'^b\right), \text{ with } y' = 0 \tag{2}$$

where $\mathbf{K}$ and $\tilde{\mathbf{K}}$ are

$$\mathbf{K} = \mathbf{P^b}\mathbf{H}^\mathrm{T}\left(\mathbf{H}\mathbf{P^b}\mathbf{H}^\mathrm{T} + \mathbf{R}\right)^{-1} \tag{3}$$

$$\tilde{\mathbf{K}} = \mathbf{P^b}\mathbf{H}^\mathrm{T}\left(\left(\sqrt{\mathbf{H}\mathbf{P^b}\mathbf{H}^\mathrm{T} + \mathbf{R}}\right)^{-1}\right)^T \times \left(\sqrt{\mathbf{H}\mathbf{P^b}\mathbf{H}^\mathrm{T} + \mathbf{R}} + \sqrt{\mathbf{R}}\right)^{-1} \tag{4}$$

The background state vector ($x^b$) contains the variables of interest from CCC400 (Table 1). In the EKF, the length of the assimilation window is 6 month (October-March and April-September), which were adapted to the southern and northern hemispheric growing seasons to effectively incorporate the proxy records stored in trees. Hence $x^b$ contains the variables of
6 months (October–March, April–September). $x^a$ stands for the analysis state vector. $\mathbf{H}$ is the forward operator that maps the



model state to the observation space (here, it is linear). $\mathbf{H}$ differs depending on the type of observation being assimilated (see Franke et al., 2017). $\boldsymbol{y}$ represents the observations. $\mathbf{K}$ is the Kalman gain matrix, and $\tilde{\mathbf{K}}$ is the reduced Kalman gain matrix. $\mathbf{P^b}$ is the background-error covariance matrix, estimated from the 30 ensemble members. A common assumption is to treat the observation-error covariance matrix ($\mathbf{R}$) as a diagonal matrix: it is presumed that the elements of $\mathbf{R}$ are uncorrelated. Therefore,

the observations can be processed serially. We set the error variances of instrumental temperature observations to 0.9 K$^2$, and of instrumental pressure data to 10 hPa$^2$. The defined error variance of documentary temperature data is 0.25 $\sigma^2$, while the errors of tree-ring proxy data are calculated as the variance of the multiple regression residuals. The assimilation is conducted on the anomaly level: we subtract both from model and from observational data their 71-yr running mean in order to deal with the biases related to systematic model errors and inconsistent low-frequency variability in the paleodata.

The use of DA in an offline manner is typical in paleoclimate reconstructions (e.g., Dee et al., 2016). Bhend et al. (2012) argue that the assimilation step is too long for initial conditions to matter, whereas there is some predictability from the boundary conditions. In addition, Matsikaris et al. (2015) found that both online and offline DA methods perform similarly in their paleoclimate reconstruction setup. Furthermore, the offline DA is advantageous as it allows using the pre-computed simulations. In our case, we can use CCC400 (Bhend et al., 2012) and test the method without having to repeat the simulations.

## 15 2.4   Spatial localization

As $\mathbf{R}$ is a diagonal matrix the EKF can be used to assimilate the observations one by one. This serial implementation makes the calculation of $\mathbf{P^b}$ simpler. $\mathbf{H}$ is then a vector (not a matrix) of the same length as $\boldsymbol{x^b}$. It is zero everywhere except for few elements (those required to model the observation). This translates to only a few columns of $\mathbf{P^b}$ that are actually required. $\mathbf{HP^bH^T}$ and $\mathbf{R}$ are then scalars (Whitaker and Hamill, 2002). This procedure also makes the localization simpler, as it needs

to be applied only to those columns. In the original setup the elements of $\mathbf{P^b}$ were Schur-product with a distance-dependent function (see Eq. (7) in Franke et al., 2017). For all the variables in the state vector, the same Gaussian function was used but with different localization length scale parameters (Table 1). For the cross-covariances between two variables, the smaller localization length scale of the two variables is applied. With the serial implementation, the calculation and localization of $\mathbf{P^b}$ is significantly simplified.

## 25 3   Experiment design

Franke et al. (2017) produced a monthly global paleoclimatological data set by using the EKF method. We leave most of the original setup unchanged and mainly focus on the estimation of $\mathbf{P^b}$. To investigate the performance of the EKF some aspects involving localization and estimation of the $\mathbf{P^b}$ matrix were tested. An overview of all experiments conducted in this study is given in Table 2. The results of the various experiments are evaluated in terms of performance measures, which then compared

to those obtained with the original setup.

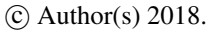



## 3.1 Spatial localization

In most of the studies, localization function is implemented in an isotropic manner. In the original setup, the same horizontally isotropic localization function was used with different localization parameters. However, such spatial symmetries may not be realistic. In the real atmosphere, correlation lengths might be longer in the zonal than in the meridional direction, due to the

prevailing winds and the weaker large-scale temperature gradients in this direction. Hence, instead of using a circular Gaussian function, we conducted an experiment with a spatially anisotropic localization function

$$C = \exp\left(-\frac{1}{2}\left(\frac{d_z^2}{L_z^2} + \frac{d_m^2}{L_m^2}\right)\right), \tag{5}$$

where $d_z$ and $d_m$ are the distances from the selected grid box in the zonal and meridional directions, respectively. $L_z$ and $L_m$ are the length scale parameters used for localization in the zonal and meridional directions, respectively. As a first experiment

we tested a 2:1 ratio for $L_z$:$L_m$. We used the values from Table 1 in the meridional direction and doubled them in the zonal direction. Thus, the resulting localization function has an elliptical shape.

## 3.2 Inflation techniques

Covariance inflation techniques are another possible method to compensate for errors in the DA system (Whitaker et al., 2008). The multiplicative inflation technique uses a small factor $\gamma$ ($\gamma > 1$) with which the $x'^b$ is multiplied (Anderson and Anderson,

1999). This type of covariance inflation accounts for filter divergence due to sampling error (Whitaker and Hamill, 2002), but can be also applied to take into account system errors (Whitaker et al., 2008). We conducted some experiments using multiplicative inflation, although in our offline approach, filter divergence is not the main concern as $\mathbf{P^b}$ is not propagated in time.

The other methodology that we adapt, shows similarities with additive inflation technique (e.g., Houtekamer and Mitchell,

2005) and with hybrid DA scheme (e.g., Clayton et al., 2013). In both methods $\mathbf{P^b}$ is modified by either adding model error (Houtekamer and Mitchell, 2005) or a so-called climatological covariance matrix (Clayton et al., 2013) to $\mathbf{P^b}$. This has given rise to the idea of generating a climatological ensemble in order to alleviate the effect of the small ensemble size. In the original setup $\mathbf{P^b}$ is approximated from only 30 members. Here, we additionally build a climatological state vector ($x^{clim}$) from randomly selected ensemble members from our 400-year long model simulation. The number of ensemble members

should be higher than the original ensemble size, but still computationally affordable. We tested numbers between 100 and 500. From $x^{clim}$ a climatological background-error covariance matrix ($\mathbf{P^{clim}}$) can be obtained. The background-error covariance matrix used in the blending experiments ($\mathbf{P^{blend}}$) is built as a linear combination of the sample covariance matrix ($\mathbf{P^b}$) and the climatological covariance matrix ($\mathbf{P^{clim}}$):

$$\mathbf{P^{blend}}\mathbf{H}^T = \beta_1 \mathbf{P^b}\mathbf{H}^T + \beta_2 \mathbf{P^{clim}}\mathbf{H}^T, \tag{6}$$

where $\beta_1$, $\beta_2$ mean the weights given to the covariance matrices. The sum of the weights is unity.



Figure 2 shows the main steps of the blending assimilation process. First, the covariance matrices were localized separately, then we blended them according to the given weights. We conducted several experiments to tune the ratio between the two covariance matrices while using different localization length scale parameters (L) (Table 2).

Since observations are assimilated serially, we also update $x^{clim}$ after an observation is assimilated with the same Kalman gain matrices as $x^b$. Thus, in the assimilation process we propagate 30 + n ensemble members, which leads to an increased computational time.

### 3.3 Temporal localization

Localizing observations in time is a special feature of the EKF due to its 6-month assimilation window. Having the state vector in half-year format, every month within the October–March or April–September time window is updated by each single observation. To test whether the covariances between a single observation and the multivariate climate fields are correctly captured, we ran an instrumental-only experiment with temporal localization. We set covariances between different months to zero.

### 3.4 Skill scores

The EKF method is tested with different localization functions and with a set of mixed background-error covariance matrices as described above. We have performed the experiments by assimilating either only proxy records (proxy-only) or only instrumental data (instrumental-only). The proxy-only experiments were carried out between 1902 and 1959, because many proxy records already end in the 1960s, while the instrumental-only experiments were tested over the 1902–2002 period. We separated the different observation types to see whether different settings perform better depending on the type of data being assimilated. We do not compare proxy-only results with instrumental-only results, hence the difference in time periods used does not matter; we simply use the longest possible time period. To evaluate the reconstructions we examined two verification measures: correlation coefficient, and reduction of error (RE) skill score (Cook et al., 1994). We use the CRU TS 3.10 dataset (Harris et al., 2014) for reference in the validation process. The presented verification measures are functions of time. Correlation is calculated between the absolute values of the ensemble mean of the analysis and the reference series at each grid point. The RE compares the reconstruction with a no knowledge prediction (such as a climatology), both expressed as deviations from a reference.

$$RE = 1 - \frac{\sum (x_i^u - x_i^{ref})^2}{\sum (x_i^f - x_i^{ref})^2} \tag{7}$$

where $x^u$ is the ensemble mean of the analysis, $x^f$ is the ensemble mean of the model background state, $x^{ref}$ is the reference dataset and $i$ refers to the time step. The RE skill scores are computed based on anomalies with respect to the 71-year running climatologies. Note that $x^f$ comes from a forced model simulation, therefore it already has skill compared with a climatological state vector. The RE is 1 if the $x^u$ is equal to $x^{ref}$. Negative RE values indicates that the background state is closer to the reference series than the analysis.





In the next section, we will focus on analysing the result of the experiments mainly over the extratropical Northern hemisphere (ENH), because most of the data are located in this region. The skill scores refer to seasonal averages of the ensemble mean.

## 4 Results and Discussions

### 4.1 Localization function

#### 4.1.1 Results

We compared the original setup applying isotropic localization function and the experiment in which an anisotropic localization function was used, to test whether we can obtain a more skilful reconstruction by implementing anisotropic localization method. As an example of the spatial reconstruction skill, we show the RE values of temperature (Fig. 3). The figures reveal that the type of localization function only resulted in small differences in both experiments. Nonetheless, there are larger areas of negative RE values (Greenland, Siberia) with the anisotropic localization function in the proxy-only experiment (Fig. 3f). In the instrumental-only experiment the decrease of RE values occur in the northern high latitudes and in the Tibetan plateau in both seasons (Fig. 3d, Fig. 3e). To have a better overview how the skill scores changed we summarize their distributions with the help of box plots. Figure 4 shows how the correlation coefficients of the three variables (temperature, precipitation and sea-level pressure) were affected in the ENH region by using the anisotropic localization function. In the instrumental-only experiment correlation values of temperature and sea-level pressure decreased in both season while for precipitation it remained mostly unchanged. The RE values show that the experiments with anisotropic localization function reduced the skill of the reconstructions, but the extent of the reduction varies with the variables and with the seasons (Fig. 5). In general, the same holds for the proxy-only experiment (Fig. 4, Fig. 5).

#### 4.1.2 Discussion

In a previous ozone reconstruction study, a seasonally and latitudinally varying localization method was tested which mostly positively affected the analysis (Brönnimann et al., 2013). Here, we increased the zonal distances to see if we can use the information of the observations for a larger region. However, the verification measures are shifted more to the negative direction. We assume that the degraded skill of the reconstruction is due to the choice of too long $L_z$, hence spurious correlations were not removed. Using anisotropic localization (doubling the Ls only in the zonal direction) consistently makes the reconstruction worse.



## 4.2 Inflation experiments

### 4.2.1 Results

The rank-deficiency of $\mathbf{P^b}$ is the main problem of ensemble-based DA techniques. To improve this issue we have tested different inflation methods.

Using the multiplicative inflation method, the model space covered by the ensemble is extended by being multiplied with a small factor ($\gamma$). To find the optimal $\gamma$ a set of experiment runs is required. We used $\gamma = 1.02$ and $\gamma = 1.12$ in our experiments, where only instrumental data were assimilated. We chose $\gamma$ from a range that was previously tested by Whitaker and Hamill (2002). Multiplying the deviations from the ensemble mean with $\gamma = 1.02$ in the assimilation process hardly affected the skill of the reconstruction over the ENH region (not shown). When we increased the value of $\gamma$ to 1.12, the RE values slightly

decreased (not shown). We did not carry out further experiments since based on the results randomly increasing the error in background field did not lead to improvement.

In the other set of experiments, we used $\mathbf{P^{blend}}$ in the update equation (Eq. 6). The experiments were run with using $\beta_2$ equal to 0.25, 0.50, 0.75, and 1 to estimate the $\mathbf{P^{blend}}$ (denoted 25c, 50c, etc.). Besides the varying weight given to $\mathbf{P^{clim}}$, the applied Ls on $\mathbf{P^b}$ and $\mathbf{P^{clim}}$ differed as well. Three Ls were used: No localization (termed no), applying Ls as in Table 1 (L) and doubling

these numbers (2L). Different combinations of the fraction of $\mathbf{P^{clim}}$ and Ls were termed accordingly (e.g., 50c_PbL_Pc2L).

We expect that estimating the covariances from 250 members instead of 30 leads to a more accurate background matrix. Hence, $\mathbf{P^{clim}}$ is less affected by the sampling error implying that long-range spurious correlations are less prominent, which makes localization less needed. We presume that using $\mathbf{P^{blend}}$ helps to better reconstruct areas which were characterized with lower skill score values in the original setup and to improve the estimation of unobserved climate variables. The reconstruction

skill of the blending experiments is always calculated from $x^\alpha$ (Fig. 2).

For the ENH region we present how the verification measures changed by replacing $\mathbf{P^b}$ with $\mathbf{P^{blend}}$ in the assimilation process. We conducted an experiment without localizing $\mathbf{P^{clim}}$ and using Ls from Table 1 on $\mathbf{P^b}$ in the constructionn of $\mathbf{P^{blend}}$. However, the skill of the reconstruction was largely reduced, implying that 250 members are not enough to avoid localization altogether (not shown).

Figure 6 and Figure 7 show the distribution of correlation coefficients and RE values, respectively. Depending on the variables and the data type being assimilated, different setups perform best. In case of assimilating only instrumental data, the most skilful temperature reconstruction was obtained from the 100c_PcL experiment in both seasons (Fig. 6a and b, Fig. 7a and b). Precipitation records were not assimilated, thus a reasonable estimation of the cross-variable covariances is essential. The skill of the precipitation reconstruction, in terms of correlation, is better than the forced simulation (Fig. 6d). However, the RE skill

score are rather decreased with the original setup over the ENH region (Fig. 8a and b). The settings of 75c_PbL_Pc2L experiment lead to improved analysis (Fig. 8c and d). The biggest improvement, in terms of RE skill score, was found in Europe (Fig. 8c and d). The 75c_PbL_Pc2L analysis also has higher skill in North-America, especially in the summer season (Fig. 8 d). The largest improvement in the sea-level pressure reconstruction was achieved in the 50c_PbL_Pc2L experiment (Fig. 6g





and h, Fig. 7g and h). In the proxy-only experiments, 75c_PbL_Pc2L is among the best performing experiments for all the variables (Fig. 6c, f, i; Fig. 7c, f, i).

We also investigated the effect of the ensemble size in the estimation of $\mathbf{P^{clim}}$. To test whether further improvements can be achieved by doubling the ensemble size of $x^{clim}$, we ran an experiment with the following setup: $\beta_1$ and $\beta_2$ are equally weighted, and L and 2L is applied on $\mathbf{P^b}$, $\mathbf{P^{clim}}$, respectively (Table 2). In the experiment we assimilated only instrumental data. The skill scores of $x^a$ (corr, RE) from the 500 ensemble members experiment showed no marked improvement compared with the same experiment with 250 ensemble members. An additional experiment was carried out with the same setup but using only 100 ensemble members in the construction of $x^{clim}$. The verification measures of the 50c_PbL_Pc2L_100m experiment are higher than the original one, and the distribution of the skill scores over the ENH region is very similar to what we obtain by using 250 members in $\mathbf{P^{clim}}$ for temperature and precipitation. However, the sea-level pressure fields from the 50c_PbL_Pc2L have higher skill than in the 50c_PbL_Pc2L_100m experiment (not shown).

Furthermore, we conducted two experiments in which only $x^b$ was updated after an observation was assimilated, and $x^{clim}$ was kept constant in the assimilation window. However, the ensemble members of $x^{clim}$ were randomly reselected for each year (October–September). The advantage of this setup compared to the setup described in Sect. 3.2 is that it is computationally less demanding since only the original 30 members keep being updated with the observations. In the first test, we give $\beta_2$=0.75 weight to $\mathbf{P^{clim}}$ with 2Ls. In the second test $\beta_2$=1, that is only $\mathbf{P^{clim}}$ used for updating $x^b$ and for localization the Ls in Table 1 were applied. By comparing the skill of the reconstructions without and with updating the climatological part, we see that the skill scores are higher when the climatological part is also updated with the information from the observations (Fig. 9). The only exception is the correlation values of sea-level pressure: when keeping the climatological part constant, they are slightly higher in both seasons (Fig. 9e and f). Nonetheless, by keeping the climatological part static in one assimilation window, the experiments still outperform the original reconstruction (Fig. 9).

### 4.3 Discussion

We have tested a number of configurations of the mixed covariance matrix $\mathbf{P^{blend}}$ to evaluate the effect of the sampling error. In numerical weather predication (NWP) applications, various methods have been designed to better estimate the errors of the background state. In hybrid DA systems, the advantages of variational and ensemble Kalman filter techniques are combined (Hamill and Snyder, 2000; Lorenc, 2003). In another method, the background-error covariances are obtained from an ensemble of assimilation experiments performed by a variational assimilation system (Pereira and Berre, 2006). In an additive inflation experiment, a term is added to the $x^a$ to account for the errors of the DA system (Whitaker et al., 2008).

In our implementation, $\mathbf{P^{blend}}$ is calculated from $x^b$ and $x^{clim}$. Using $\mathbf{P^{blend}}$ in the assimilation process improved on the reconstruction performed with the original setup. The skill scores show the largest improvement in the sea-level pressure reconstruction. Moreover, the skill of the precipitation reconstruction also improved, indicating that $\mathbf{P^{clim}}$ helps to better estimate the cross-covariances of the background errors between the variables. In general, increasing the weight of $\mathbf{P^{clim}}$ in forming $\mathbf{P^{blend}}$, positively affected the skill of the analysis. The 100c_PcL experiment, in which $\mathbf{P^{blend}}$ is equal to $\mathbf{P^{clim}}$, is similar to the DA technique used in the last millennium climate reanalysis (LMR) project (Hakim et al., 2016). In the LMR, 100



randomly chosen ensemble members form a climatological state vector, which is used in each assimilation window and is updated with the observations. In this study, $x^{clim}$ is randomly resampled every year and primarily used in the estimation of $\mathbf{P^{blend}}$. The analysis of the 100c_PcL experiment is more skilful, than the original reconstruction. The settings used in the 100c_PcL experiment lead to the best temperature reconstruction when only instrumental measurements are assimilated. How-

ever, other settings performed better for different variables and observation types. By applying no localization on $\mathbf{P^{clim}}$ in the 50c_PbL_PcnoL experiment we obtained a less skilful reconstruction than by using the other two localization schemes. The skills reduced especially over the areas where no local observations were assimilated. Using 2Ls for localizing the covariances of $\mathbf{P^{clim}}$ in the instrumental-only experiments resulted in better analysis of sea-level pressure (50c_PbL_Pc2L) and helped to better reconstruct summer precipitation. Among the proxy-only experiments, 75c_PbL_Pc2L shows the best skill for pressure

reconstruction. Here, pressure data are not assimilated, and the result suggests that by applying longer Ls, the cross-variable covariances are better treated. The results of the experiments show that with a mixed covariance matrix implementation a major drawback of the ensemble-based DA system, due to the limited ensemble size, can be improved.

### 4.4 Localization in time

#### 4.4.1 Results

Since six monthly time steps were combined in one state vector (one assimilation window), covariances between different months also need to be considered. An additional experiment was conducted in which the (localized) $\mathbf{P^b}$ was multiplied with a temporal localization function when instrumental data were assimilated. This is a specific experiment due to the structure of EKF. The assimilation window in the EKF is 6-month, hence a single observation is enabled to adjust all the meteorological variables in $x^b$ in a half-year time window. In the temporal localization experiment, the information from a given observation

can only modify the different climate fields in its current month, while leaving all other fields of the 5 months unchanged (Table 2). In general, the skill scores indicate an improvement. The difference of RE values between the temp_loc and original experiments are mostly positive over the northern high latitude areas (Fig. 10).

#### 4.4.2 Discussion

The higher skill scores with temporal localization (Fig. 10) indicate that the cross-covariances in time were not correctly

represented by $\mathbf{P^b}$. Hence, it is likely that in the original setup some non-physical covariances were taken into account. Applying the same assimilation scheme to another problem (estimating the two-dimensional ozone distribution from an ensemble of chemistry-climate models and historical observations), Brönnimann et al. (2013) used a localization time scale of 3 months based on empirical studies.



## 5 Conclusions

In this study, an offline data assimilation approach was used to test the effect of the estimation of the background-error co-
variance matrix in a climate reconstruction. Several experiments were evaluated with different verification measures to see
which background-error covariance matrix estimation techniques improve the skill of the reconstruction. The validation of
5    the presented techniques suggests the following: 1. Applying an anisotropic localization function on the sample covariance
matrix did not improve the reconstruction; 2 Constructing the background-error covariance matrix from the sample and clima-
tological covariance matrices, allows using longer localization length scales, and it leads to higher skill scores; 3. Assimilating
early instrumental data with temporal localization leads to a better analysis. To which extent the different techniques helped
in the estimation of the background-error covariance matrix varies geographically and also depends on the climate variable
10   being reconstructed. The cross-variable covariances of the background-error covariance matrix can provide information from
unobserved climate variables. Including climatological information in the estimation of precipitation has lead to a better re-
construction, especially in Europe. Estimating sea-level pressure with the blended $\mathbf{P^{blend}}$ matrix also improved the skill of
the reconstruction. For instance, the 50c_PbL_Pc2L experiment performs constantly better than the original setup. This study
shows that results can be improved by better specifying the background-error covariance matrix. In the future we will combine
15   all the techniques that lead to more skilful analyses to produce a climate reconstruction over the last 400 years.

*Competing interests.* The authors declare that they have no conflict of interest.

*Acknowledgements.* The study was funded by the Swiss National Science Foundation (project 162668) and by the European Union (H2020 / ERC
grant number 787574 PALAEO-RA).



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





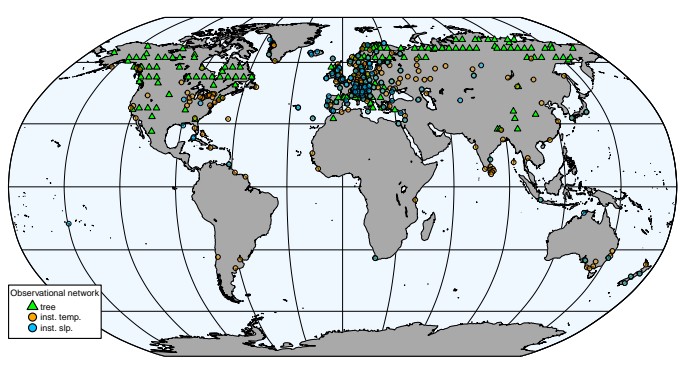

**Figure 1.** The observational network in 1904, before quality check.





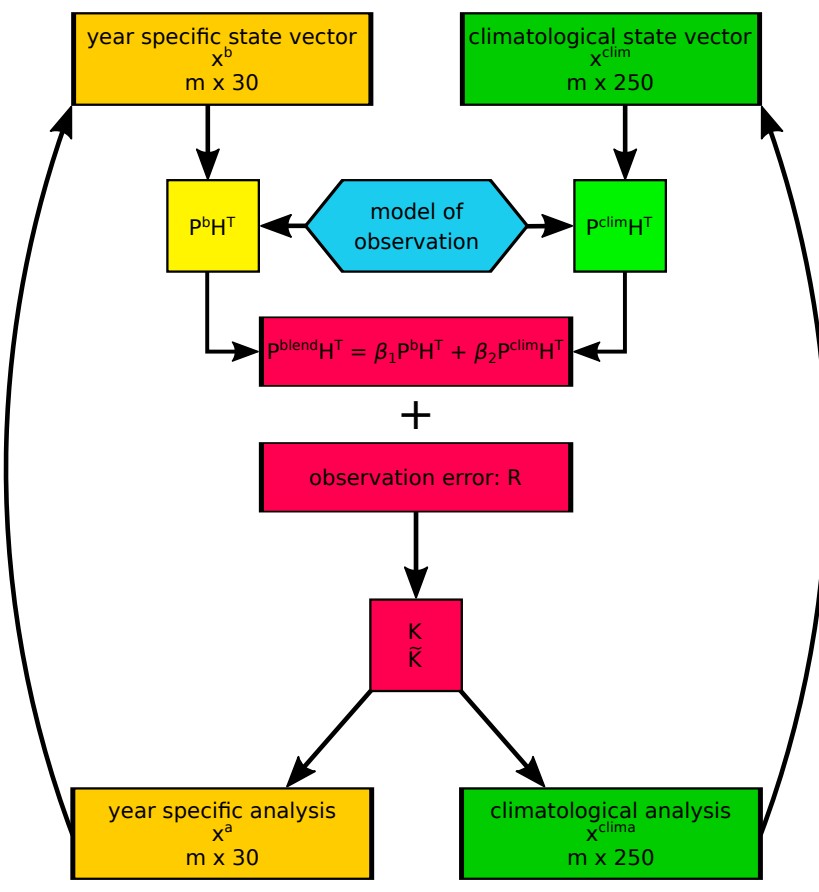

**Figure 2.** The main steps of the blending experiment in one assimilation window. The blended covariance matrix $\mathbf{P^{blend}}$ is calculated as a linear combination from the year specific and climatological covariance matrices. The calculation of the Kalman gain ($\mathbf{K}$) and reduced Kalman gain ($\mathbf{\tilde{K}}$) matrices is the same as in Eq.3 and Eq. 4 except the covariance matrix is replaced with $\mathbf{P^{blend}}$. The observation is assimilated to both state vectors and these analysis become to the starting point for assimilating the next observation.





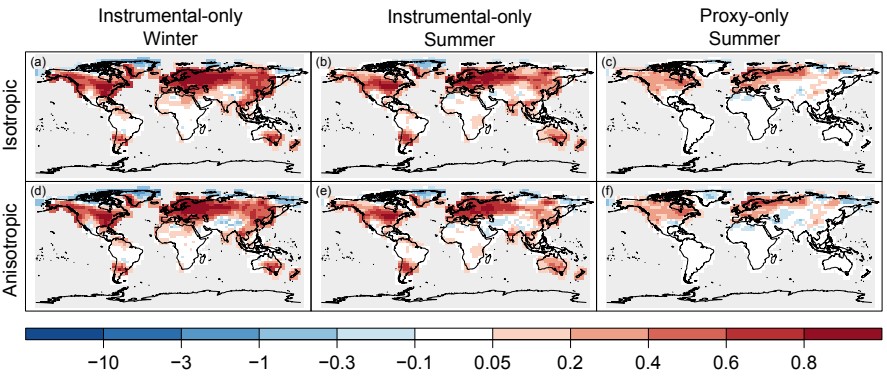

**Figure 3.** Spatial skill of temperature reconstruction presented by RE values, assimilating only instrumental data (a,b,d,e) and only proxy records (c,f). Comparing the skill of the reconstruction using isotropic localization function (a,b,c) versus an anisotropic localization function (d,e,f). Panel a and d show the skill in the winter season, while panel b, c, e, and f illustrate the skill in the summer season.





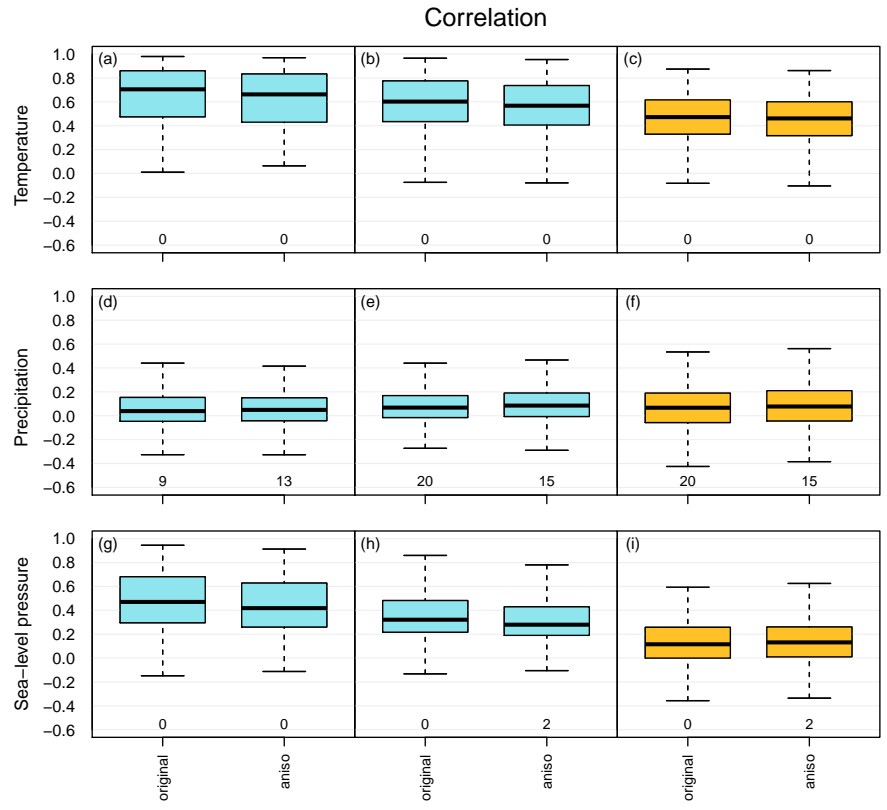

**Figure 4.** Distribution of the correlation coefficient values in the ENH region in the winter (left column) and summer (middle and right columns) half-years of temperature (a,b,c), precipitation (d,e,f) and sea-level pressure (g,h,i). Blue is the instrumental-only experiment and yellow is the proxy-only experiment. The midline of the box is the median. The lower (upper) border of the box is the first (third) quartile. The whiskers extend up to 1.5 times the interquartile range; beyond these distances the number of outliers are given under the box plots. The grid boxes were not area-weighted.





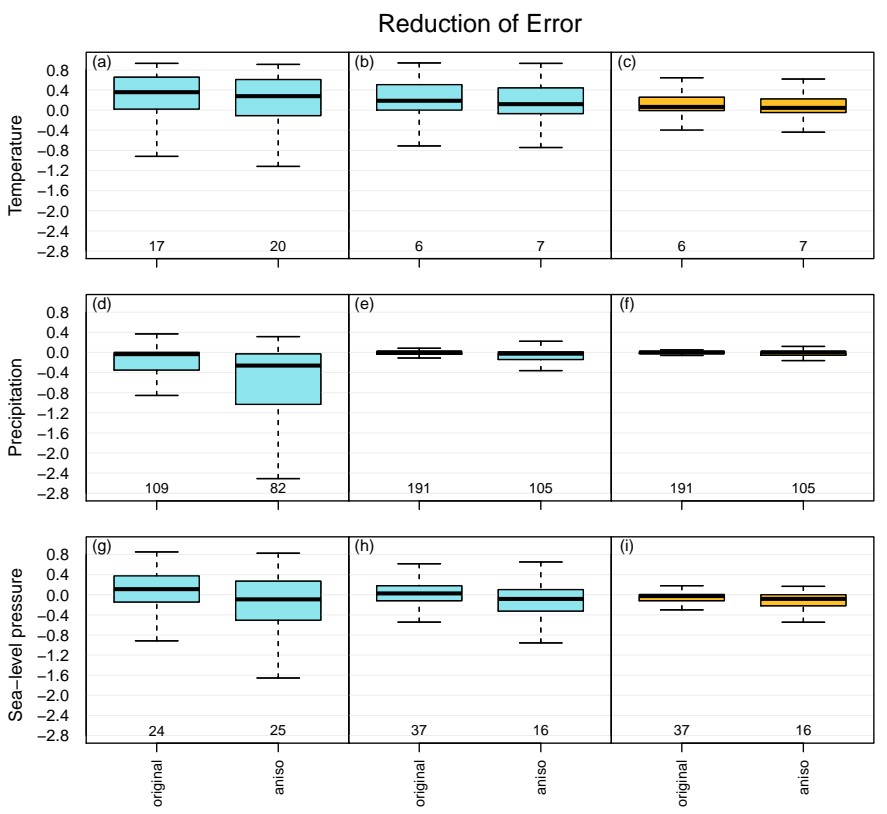

**Figure 5.** Distribution of RE values, as in Fig. 4.





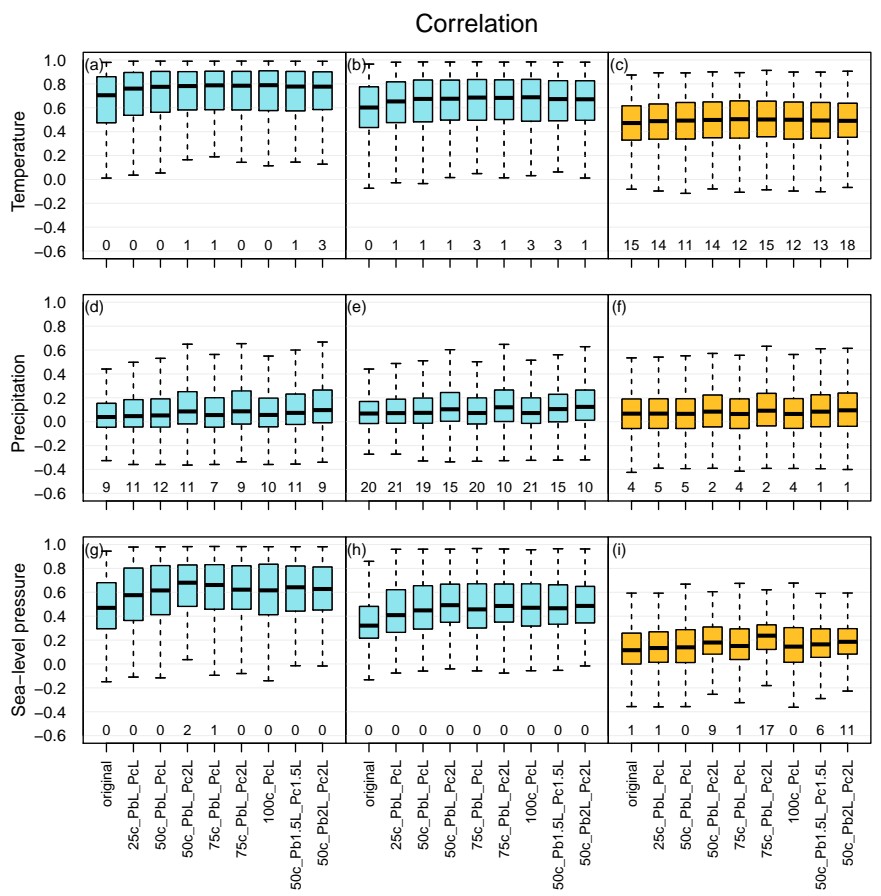

**Figure 6.** Distribution of correlation coefficients in the different mixed background-error covariance matrix experiments in the ENH region. The left column shows the skill of the reconstruction in the winter seasons, while the middle and right columns in the summer season. The labels on the x-axis indicating the experiments. Box plot, number on the panels and colors represent the same as in Fig. 4.





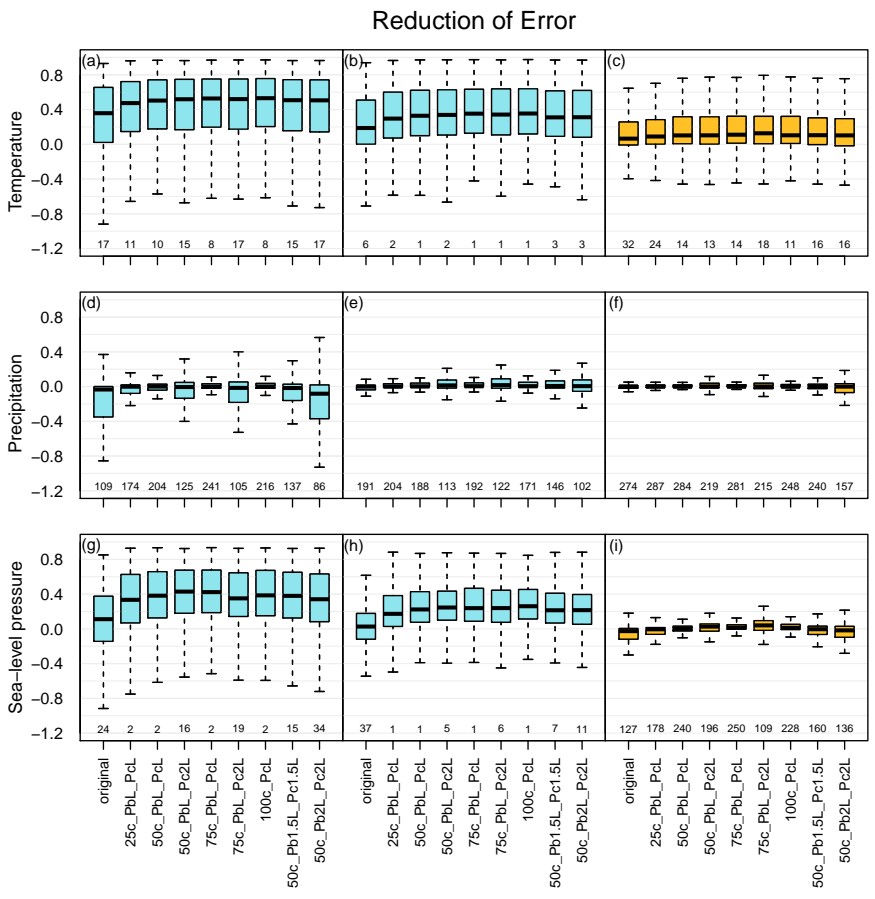

**Figure 7.** Distribution of RE values in the different mixed background-error covariance matrix experiments in the ENH region, otherwise same as in Fig. 6.





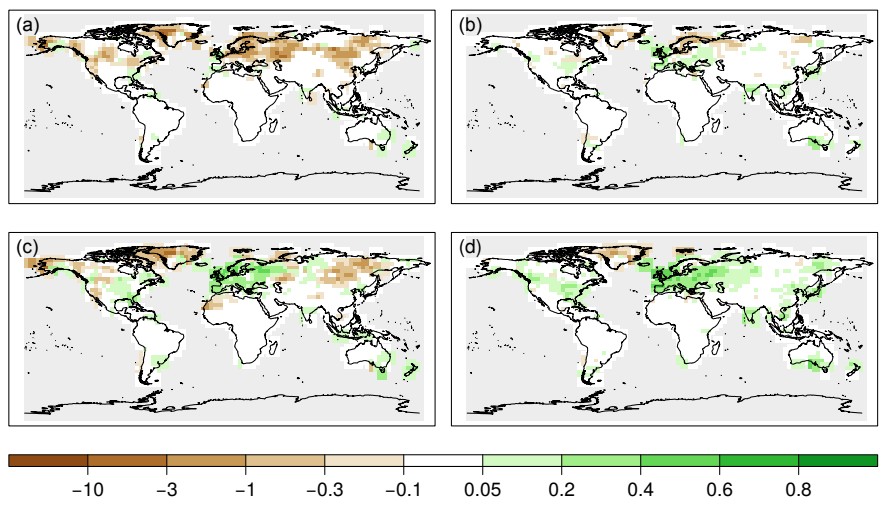

**Figure 8.** Spatial reconstruction skill of precipitation in terms of RE values. Panel a and b show the skill using the original setup, and panel c and d show the result of the 75c_PbL_PcL2 experiment. The skill in the winter season presented in the a and c panel and for summer on the b and d panels.





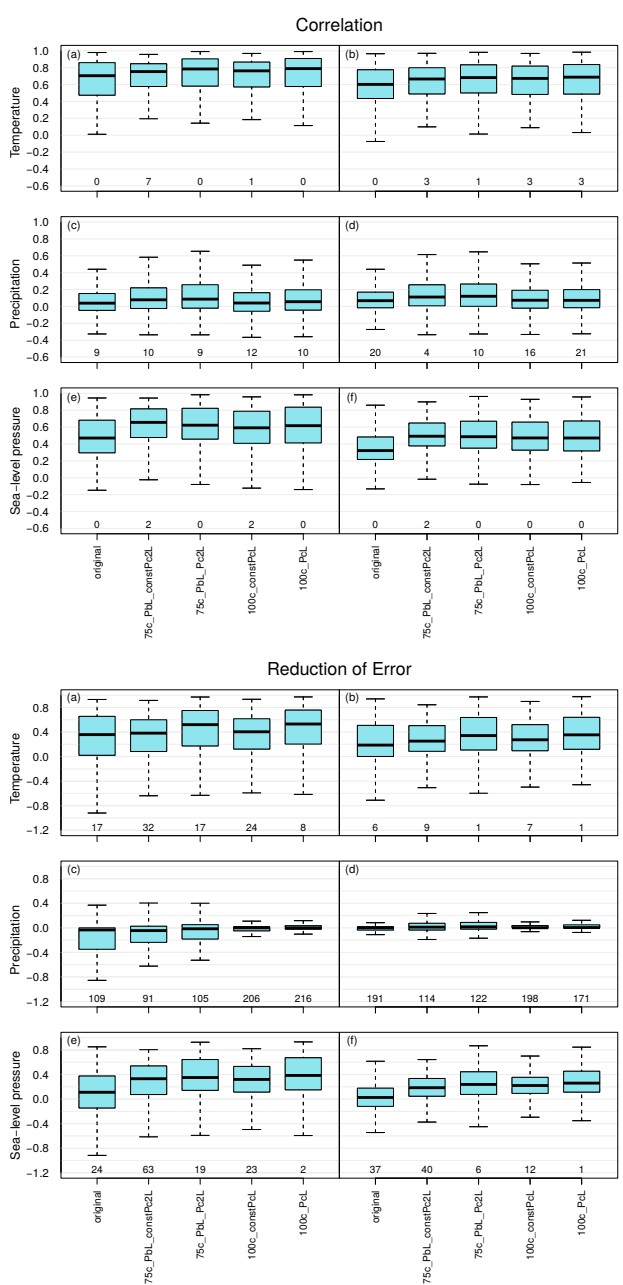

**Figure 9.** Distribution of skill scores over the ENH region. The skill of the original setup is compared with experiment 75c_PbL_constPc2L, 75c_PbL_Pc2L, 100c_constPcL, and 100c_PcL. Distribution of correlation coefficients in the winter (left column) and in the summer (right column) seasons. Distribution of RE values in the winter (left column) and in the summer (right column) seasons.





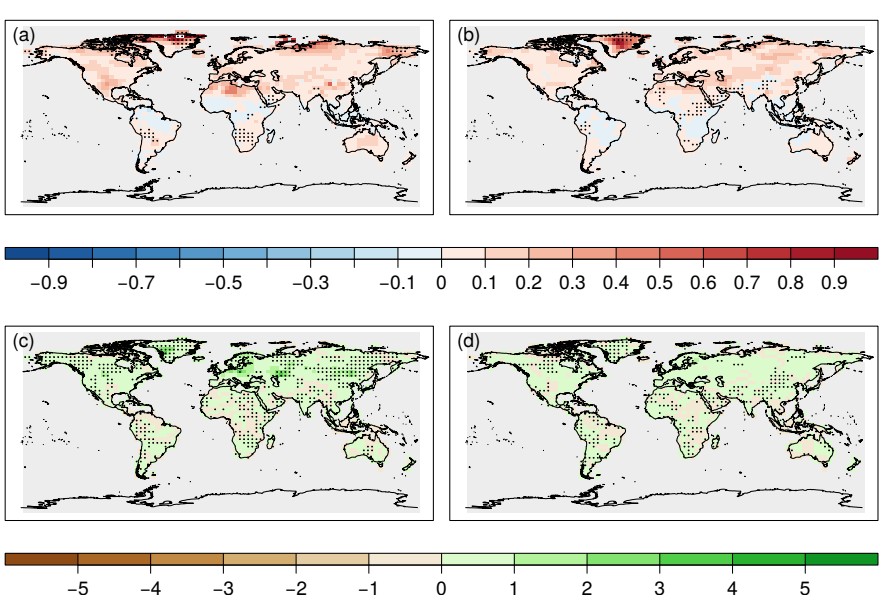

**Figure 10.** Difference of the RE skill between the temporally localized experiment and the original setup: temperature (a) in winter and (b) in summer; precipitation (c) in winter and (d) in summer. The black dots indicate the negative RE values in the temporally localized experiment.



**Table 1.** Defined localization length scale parameters

| Variable | Localization length scale (km) |
|---|---|
| Temperature (2m) | 1500 |
| Precipitation | 450 |
| Sea-level pressure | 2700 |



**Table 2.** Summary of the experiments carried out in this study. The name of the experiments indicate which settings were used in the assimilation. Localization refers to the shape of the localization function applied on $P^b$. $\gamma$ is the multiplicative inflation factor. $x^{clim}$ indicate from how many ensemble members the climatological state vector was constructed. $x^{clim}$const stands for keeping the climatological part in the blending experiment unchanged in one October–September time window. $P^b$loc indicates the localization length scale parameter applied for localizing $P^b$. $\beta_2$ refers to the weight given to $P^{clim}$. $P^{clim}$loc indicates the localization length scale parameter applied for localizing $P^{clim}$. i and p stands for instrumental-only and proxy-only observations experiments, respectively.

| Name | Localization | $\gamma$ | Blending | | | | | Temporal localization | Obs. type |
|---|---|---|---|---|---|---|---|---|---|
| | | | $x^{clim}$ | $x^{clim}$const | $P^b$loc | $\beta_2$ (%) | $P^{clim}$loc | | |
| original | iso | no | | | | | | no | i,p |
| aniso | aniso | no | | | | | | no | i,p |
| mul1.02 | iso | 1.02 | | | | | | no | i |
| mul1.12 | iso | 1.12 | | | | | | no | i |
| 25c_PbL_PcL | iso | no | 250 | no | L | 25 | L | no | i,p |
| 50c_PbL_PcnoL | iso | no | 250 | no | L | 50 | no | no | i |
| 50c_PbL_PcL | iso | no | 250 | no | L | 50 | L | no | i,p |
| 50c_PbL_Pc2L_100m | iso | no | 100 | no | L | 50 | 2L | no | i |
| 50c_PbL_Pc2L | iso | no | 250 | no | L | 50 | 2L | no | i,p |
| 50c_PbL_Pc2L_500m | iso | no | 500 | no | L | 50 | 2L | no | i |
| 50c_Pb1.5L_Pc1.5L | iso | no | 250 | no | 1.5L | 50 | 1.5L | no | i,p |
| 50c_Pb2L_Pc2L | iso | no | 250 | no | 2L | 50 | 2L | no | i,p |
| 75c_PbL_PcL | iso | no | 250 | no | L | 75 | L | no | i,p |
| 75c_PbL_Pc2L | iso | no | 250 | no | L | 75 | 2L | no | i,p |
| 75c_PbL_constPc2L | iso | no | 250 | yes | L | 75 | 2L | no | i |
| 100c_PcL | iso | no | 250 | no | | 100 | L | no | i,p |
| 100c_constPcL | iso | no | 250 | yes | | 100 | L | no | i |
| temp_loc | iso | no | | | | | | yes | i |