# Peer review of "Impact of different estimations of the background-error covariance matrix on climate reconstructions based on data assimilation"

_Climate of the Past, 2018_

## Referee Comment (RC1) · Anonymous Referee #1 · 4 Feb 2019

Review of CP-2018-168

This paper tests several different methodological choices that are typically made (or could be made) in paleoclimate reconstructions using DA. I think this presents a good and valuable presentation and discussion of these choices. The findings and suggestions for future reconstructions are very helpful to the community performing these types of reconstructions.

Section 3.1: Is there any specific justification for the choice of the ratio of L_z and L_m being 2:1? Could this, or some other ratio, be justified by looking at the correlation length scale in observational data?

[Figure]

Section 3.1: Is there any justification for the specific localization values that you chose for each variable that was reconstructed? Are these values data-driven or just educated guesses? Were any experiments done to test on optimal localization value? I would assume that if these values were used based on weather DA experiments, they might not apply on the longer paleo time scales where one would generally expect the correlation length scales to be larger.

Section 4.2.1: When you are comparing the distributions, you say that for example, the most skillful reconstruction is obtained from the 100c_PcL experiment. What is the basis for saying it's the best? What aspect of the distribution are you comparing? The median or some other specific value(s)?

Many of the distributions shown in the figures look very similar so it was hard for me to feel confident about the statement that one particular set of reconstruction choices was better than another. Are the distributions statistically distinct? Instead of comparing the distributions, would it be possible to show the differences compared to the "original" reconstruction (i.e., you'd compute the difference in the skill score for each location and then summarize this distribution of differences in the plots)? I'm wondering if this, or something similar, might make the differences more clear. Because currently when I look at the distributions, many of them look very similar and perhaps even statistically indistinguishable.

Fig 8 & 10: It would be very helpful to give a little more explanatory information/labeling on each panel, such as was done in Fig 3.

---

## Referee Comment (RC2) · Anonymous Referee #2 · 12 Mar 2019

Review of

'**Impact of different estimations of the background-error covariance matrix on climate reconstructions based on data assimilation'**

by V. Valler, J. Franke, and S. Brönnimann

**Recommendation: minor revisions**

This manuscript identifies the best choices from a number of different spatial and temporal localization approaches and from different inflation techniques for the background error covariance matrix in Ensemble Kalman Filters used in paleoclimatic applications. The optimization of these technical details in data assimilation is important for the growing paleoclimate data assimilation community. The results are systematically derived and the manuscript is in general well written. I support publication after the points listed below have been clarified or corrected.

**Specific comments**

Page 1, Line 2, replace 'of the assimilation system' with 'of some assimilation systems'

Page 1, Lines 17/18, 24/25, 'boundary conditions' are specifications of state variables at the boundaries of a model domain, and thus not the same as 'forcings', which are external influences on the system. The two should be distinguished throughout the text. It seems that here the statement are about forcings. If so, reformulate avoiding the use of 'boundary condition'.

Page 2, line 6, 'linear models' of what? I think it should by 'linear dynamical systems'. A short comment on why KFs are used with non-linear systems, including GCMs, would be good. 'Gaussian distributions' of what? The state variables?

Page 2, line 13-16, I suggest using 'stationary offline' and 'transient offline' for the two approaches.

Page 2, lines 17/18, 'The true climate state is not known, therefore it has to be estimated'. Does 'it' refer to 'the true climate state', as the sentence suggests or to 'the uncertainty of the background state', which would link better to the first sentence in this paragraph? This sentence should be clarified, or it could simply be deleted (which I think is the better option).

Page 2, line 20, What is a non-simplified KF? A KF with a 'true' background error covariance matrix? If so, how can this exist? The background error covariance has always to be estimated somehow. Please clarify the statement.

Page 2, line 23, It seems that the sampling error for the background error is not only a random error, but leads to a systematic underestimation of the background error, otherwise inflation would not be a suitable approach. Please explain better.

Page 2, lines 25/26. The statement on distribution of ensemble members refers to online approaches, but the approach used by the authors is an offline approach. This is confusing. Please briefly explain how the ensembles are generated in an online KF, and that in offline approaches the ensemble is given, but that the background error covariance still needs to be inflated.

Page 3, line 14, replace 'other method' with 'additive method'

Page 3, line 25, replace ''form' with 'from'

Page 3, line 28, Don't use 'forced by boundary conditions', as forcings and boundary conditions are different (see comment above). If I understand correctly for all ensemble members the same greenhouse gas, solar and volcanic forcings have been used, as well as the same SST boundary conditions. Please clarify.

Page 3, Lines 29-31, The SST reconstruction can be expected to strongly influence the results of this data assimilation approach with an atmosphere-only GCM. There should be some comments on how the SST reconstructions have been made, what is known about their uncertainties, and why this approach is taken rather than data assimilation with a coupled atmosphere-ocean GCM.

Page 3, line 30, I think 'till' should not be used in formal writing and should be replaced with 'until' (also later in the text).

Page 4, line 1, replace 'boundary conditions' with 'forcing', and make a separate statement on land-surface boundary conditions, including which variables are prescribed.

Page 4, line 16, 'CCC400' has not been introduced

Page 4, line 23, If I understand correctly the deviations of the ensemble members from the ensemble mean are updated in the online EnSRF according to equation 2, but not in the offline EKF, which uses an existing ensemble. Please clarify.

Page 4, lines 28-30, the statements on the 6 month periods are partly redundant

Page 5, lines 5-7, How have the error variances been chosen? Should sigma^2 be K^2? If so, why is the error for documentary data smaller than for instrumental data? Which multiple regression?

Page 5, lines 16 – 19. The notation is not clean. In line 16 it is said that R is a diagonal matrix, in line 19 that R is a scalar. The problem is that the same notation is used for an equation using the full set of observations (where R is a diagonal matrix) and for the equation when the individual observations are assimilated sequentially. Please reformulate.

Page 6, line 1, replace 'localization function' with 'the localization function'.

Page 6, line 19/20, replace 'additive inflation' with 'the additive inflation', and 'hybrid' with 'the hybrid'.

Page 6, line 23-25, The explanation is confusing. One can select ensemble members for the whole period or some or all ensemble members for some timesteps; how exactly are the climatological state vector and the associated error covariance matrix calculated? The simulations have already

been performed; why are there substantial computational costs for using a large number of ensemble members?

Page 6, line 29, H^T is at the end of all terms in the equation. Can it not simply be deleted?

Page 7, line 4-5. It is not clear how x^clim is calculated and updated, what n is, and what 'propagated' means in an offline assimilation scheme.

Page 8, line 7, replace 'isotropic localization' with 'an isotropic localization'.

Page 8, line 15, ENH is not defined

Page 9, line 3, Explain why P^b does not have full rank, why this is a problem, and what this has to do with inflation.

Page 9, lines 5-6. This is not well phrased; it is not the covered model space that is multiplied with the inflation factor.

Page 9, line 16, It has not yet been mentioned in the main text in section 3 that 250 members have been chosen and how they have been selected (see also earlier comment on this).

Page 12, line 2, replace 'verification' with 'validation' (this is used elsewhere in the text) or 'performance'

Page 12, line 6, missing full stop after '2'

Page 12 and also in main text, Add a comment on whether the skill differences found are substantial and practically relevant.

Page 14, line 11, 'kalman' should be upper case

Page 14, line 16/17, typos and missing spaces

Figs. 1, 3, 8, 10 should be bigger

---

## Author Comment (AC1) · 5 Apr 2019

**Reviewer #1**

**This paper tests several different methodological choices that are typically made (or could be made) in paleoclimate reconstructions using DA. I think this presents a good and valuable presentation and discussion of these choices. The findings and suggestions for future reconstructions are very helpful to the community performing these types of reconstructions.**

We thank the Reviewer for the positive comments on our work and for the suggestion how to improve our figures.

**Section 3.1: Is there any specific justification for the choice of the ratio of L_z and L_m being 2:1? Could this, or some other ratio, be justified by looking at the correlation length scale in observational data?**

> The idea behind a longer length scale in zonal direction than in meridional direction is based on the zonal flow in the atmosphere. On multi-annual to multi-decadal time scales multiple processes act in meridional direction, e.g. a widening/shrinking of the Hadley cell, shifts of the ITCZ or changes in atmospheric modes like AMO or NAO. These can shift the the zonal circulation northward or southward but the zonal coherence will be less effected. That is the reason why we had the hypothesis that may have longer decorrelation distances in zonal direction. We will explain this hypothesis in the revised version of the manuscript.

**Section 3.1: Is there any justification for the specific localization values that you chose for each variable that was reconstructed? Are these values data-driven or just educated guesses? Were any experiments done to test on optimal localization value? I would assume that if these values were used based on weather DA experiments, they might not apply on the longer paleo time scales where one would generally expect the correlation length scales to be larger.**

> The localization length scale parameters were defined based on the spatial correlation of the variables in the monthly ECHAM model simulation fields. In Section 2.4 we refer to the paper by Franke et al. 2017 how the localization was done in the original setup. We used the same localization length scale parameters for localizing the sample covariance in most of our experiments to evaluate improvements in comparison with this initial setup. For this study, we calculated the latitudinal dependency of correlation of the state variables from a bigger ensemble of the model than in Franke et al (2017). The result suggested that the longer length scale parameters can be applied in the tropics and the predefined length scale parameter of precipitation is probably too strict. Based on the rather strict decorrelation length scale in the previous study and the assumption that the covariances can be better estimated from a bigger ensemble, we used doubled length scale parameters in some of the experiments for localizing the climatological covariances. In this case, the L for temperature is 3000 km, which means that the correlation is decreased close to zero approximately 6000 km away from the observation. We did not carry out further experiment to find the optimal localization value because even double the localization distance hardly changed the reconstruction skill. Hence, our system does not appear to be very sensitive on the localization distance as long as it remains in a reasonable range. On the one hand, we do not further restrict the localization because that would limit updates to a small regions around observations. On the other hand, our experiments without localization showed negative reconstruction skill in locations far away from observations, even with the error covariance matrix is calculated from climatology. We will provide this additional explanation in the revised manuscript.

**Section 4.2.1: When you are comparing the distributions, you say that for example,the most skillful reconstruction is obtained from the 100c_PcL experiment. What is the basis for saying it's the best? What aspect of the distribution are you comparing? The median or some other specific value(s)?**

Yes, for comparison we used only the median.

**Many of the distributions shown in the figures look very similar so it was hard for me to feel confident about the statement that one particular set of reconstruction choices was better than another. Are the distributions statistically distinct?**

We agree with the Reviewer that the distributions of the skill of the experiments over the extratropical Northern hemisphere look similar. We have not checked whether the distributions are statistically distinct. In the revised paper we will provide some statistical evaluation of the experiments.

**Instead of comparing the distributions, would it be possible to show the differences compared to the "original"reconstruction (i.e., you'd compute the difference in the skill score for each location and then summarize this distribution of differences in the plots)? I'm wondering if this,or something similar, might make the differences more clear. Because currently when I look at the distributions, many of them look very similar and perhaps even statistically indistinguishable.**

Thank you for your suggestion. We will make the plots as it was suggested and if the differences become more distinguishable we will replace the original figures (Fig. 4-7 and 9).

**Fig 8 & 10: It would be very helpful to give a little more explanatory information/labeling on each panel, such as was done in Fig 3.**

We will add more labels to the figures.

---

## Author Comment (AC2) · 5 Apr 2019

**Reviewer #2**

**This manuscript identifies the best choices from a number of different spatial and temporal localization approaches and from different inflation techniques for the background error covariance matrix in Ensemble Kalman Filters used in paleoclimatic applications. The optimization of these technical details in data assimilation is important for the growing paleoclimate data assimilation community. The results are systematically derived and the manuscript is in general well written. I support publication after the points listed below have been clarified or corrected.**

We thank the Reviewer for the careful revision of the manuscript. We will follow his/her recommendations and add further details at the indicated points to provide better understanding how ensemble Kalman filter methods work, how we set up the blending experiment and how covariance inflation techniques can help to better estimate the background-error covariance matrix. We are also thankful for improving the English of the manuscript.

**Specific comments**

**Page 1, Line 2, replace 'of the assimilation system' with 'of some assimilation systems'**

    We will replace it.

**Page 1, Lines 17/18, 24/25, 'boundary conditions' are specifications of state variables at the boundaries of a model domain, and thus not the same as 'forcings', which are external influences on the system. The two should be distinguished throughout the text. It seems that here the statement are about forcings. If so, reformulate avoiding the use of 'boundary condition'.**

    Will be replaced as:

    Climate models constrained with realistic, time-dependent external forcings provide fields that are consistent with these forcings and the model physics.

    If information from observations is optimally blended with climate model simulations, the result is the best estimate of the climatic state, given the observations, given the external forcings, and given the known climate physics.

**Page 2, line 6, 'linear models' of what? I think it should by 'linear dynamical systems'. A short comment on why KFs are used with non-linear systems, including GCMs, would be good. 'Gaussian distributions' of what? The state variables?**

    Yes, the KF is optimal for linear systems (linear dynamical model and linear observation operator) with Gaussian error distributions (model, background, and observation). In atmospheric data assimilation the most commonly used KFs are the ensemble-based Kalman filters that can handle some non-linearity in the dynamics and in the observation operator. Moreover, in our approach we use an offline implementation of the ensemble square root filter, i.e. we only adjust the precomputed model simulations with the observations and never deal with the question of model dynamics. Nevertheless, we will remove this sentence because it is not necessary here and just caused confusion.

**Page 2, line 13-16, I suggest using 'stationary offline' and 'transient offline' for the two approaches.**

Thank you for the suggestion. We will use the terms "forcing-independent offline" for "stationary offline" and "forcing-dependent offline" for "transient offline" to better distinguish between the two methods.

**Page 2, lines 17/18, 'The true climate state is not known, therefore it has to be estimated'. Does 'it' refer to 'the true climate state', as the sentence suggests or to 'the uncertainty of the background state', which would link better to the first sentence in this paragraph? This sentence should be clarified, or it could simply be deleted (which I think is the better option).**

We will delete the sentence.

**Page 2, line 20, What is a non-simplified KF? A KF with a 'true' background error covariance matrix? If so, how can this exist? The background error covariance has always to be estimated somehow. Please clarify the statement.**

We will replace 'Ensemble-based KFs are simplification of the KF' to 'Ensemble-based KFs are approximation of the KF'.

In the Kalman filter, the background-error covariance matrix is given as

$$P^b = \overline{(x^b - x^t)(x^b - x^t)^T}$$

the superscripts b and t indicate the background state and the true state, while the overline marks the expectation value.

In practice, as was said, the background-error covariance matrix has to be estimated. In ensemble-based Kalman filter techniques, the ensemble covariance will provide the background-error covariance matrix and the true state is estimated by the ensemble mean.

$$P^b_e = \overline{(x^b - \overline{x^b})(x^b - \overline{x^b})^T}$$

**Page 2, line 23, It seems that the sampling error for the background error is not only a random error, but leads to a systematic underestimation of the background error, otherwise inflation would not be a suitable approach. Please explain better.**

Ensemble-based filters with flow-dependent covariances can diverge due to sampling error. Our small 30 member ensemble may result in an error underestimation, because we do not assess the uncertainties in SST boundary conditions and external forcings. In an online DA system if the error of the analysis is underestimated and it is forwarded to the next time step, it will result in giving less and less weight to the observations in the following time steps. This should not a problem in offline approaches. However, we conduct this experiment for completeness because as we say in the manuscript, there are two commonly used methods to reduce the negative effect of sampling error: inflation (e.g., Anderson and Anderson, 1999), and localization (e.g., Hamill et al., 2001) of the background-error covariance matrix."

**Page 2, lines 25/26. The statement on distribution of ensemble members refers to online approaches, but the approach used by the authors is an offline approach. This is confusing. Please briefly explain how the ensembles are generated in an online KF, and that in offline approaches the ensemble is given, but that the background error covariance still needs to be inflated.**

To avoid any confusion, we will change the sentence to: "Two methods are commonly used in online KF approaches to reduce the negative effect of sampling error:" How the ensembles are generated in an online KF is already described on page 2, line 8-11. This remains the same in ensemble-based KFs: the ensemble members are updated with the observation when they become available and the updated state, in case of ensemble-based KFs, the updated ensemble members and their uncertainties are propagated forward by the model to the next time step. We will add a short comment here that these steps are valid for ensemble-based KFs as well. Finally, with regard to the need to inflate the background error covariance, please see our answer to the last comment on page 2, line 23.

**Page 3, line 14, replace 'other method' with 'additive method'**

We will replace it.

**Page 3, line 25, replace ''form' with 'from'**

We will replace it.

**Page 3, line 28, Don't use 'forced by boundary conditions', as forcings and boundary conditions are different (see comment above). If I understand correctly for all ensemble members the same greenhouse gas, solar and volcanic forcings have been used, as well as the same SST boundary conditions. Please clarify.**

Yes, the same forcings and boundary conditions were used in the 30 members.

Will be replaced as:

The 30 ensemble members were forced and driven with the same external forcings and with the same boundary conditions.

**Page 3, Lines 29-31, The SST reconstruction can be expected to strongly influence the results of this data assimilation approach with an atmosphere-only GCM. There should be some comments on how the SST reconstructions have been made, what is known about their uncertainties, and why this approach is taken rather than data assimilation with a coupled atmosphere-ocean GCM.**

The main reason why we use the SST reconstruction by Mann et al. 2009 is that it was the only available reconstruction at the time when we ran the simulation ensemble, like we already explained in the paper: "For sea-surface temperatures (SSTs), which have a particularly large effect on the simulations, the reconstruction by Mann et al. (2009) was used. This is the only global gridded SST reconstruction that dates back till 1600."

We already describe limitations of the reconstruction and how we tried to improve them in the discussion paper, too: "The SST reconstruction by design captures interdecadal variations (Mann et al., 2009), hence intra-annual variability dependent on a El Niño/Southern Oscillation reconstructions (Cook et al.,2008) was added to the SST fields." We will add a reference to the paper of Bhend et al. 2012, which explains the simulations in a little more detail.

There are multiple reasons why we do not use a coupled atmosphere-ocean model:

1. We understand a paleo-reanalysis to be a product that best describes the atmospheric states of the past globally. 400 years of just externally forced coupled atmosphere-ocean

simulations would not be in close agreement with the true state of the ocean in the past and e.g. not produce El Niño events in the years when they actually occurred. Therefore such simulations would not capture related teleconnections patterns in the corresponding years, either. We cannot gain the same information that currently comes from the SST boundary conditions with our data assimilation setup because we only assimilate absolutely dated tree-ring proxies on land and mostly at mid to high latitude. It would be an option to assimilate coral data in the future but there are few records that back to 1600 and their assimilation has not been tested yet. That is why coral data assimilation in not an option for the presented sensitivity experiments. The focus of this study is to show, how the paleo data assimilation setup published by Franke et al. 2017 can be further improved without introducing further changes that would blur the effect of the improved covariance estimation.

2. If we would like the ocean to be in closer agreement with the observations in a coupled model, we would need to do an online assimilation scheme that could take the longer memory of the ocean into account whereas our atmospheric states have no memory that extends beyond a months or year when the next proxy observation becomes available. Hence, running a coupled ocean-atmosphere model would not allow us to work with our offline assimilation system and do all the sensitivity experiments, which we present in this study.

3. Finally, an ensemble of coupled simulations plus the additional spin-up time for the ocean to reach equilibrium would have simply not been computationally affordable.

**Page 3, line 30, I think 'till' should not be used in formal writing and should be replaced with 'until' (also later in the text).**

We will replace it, also in page 4, line 10-12

**Page 4, line 1, replace 'boundary conditions' with 'forcing', and make a separate statement on land-surface boundary conditions, including which variables are prescribed.**

Will be replaced as:

Further forcings include solar irradiance, volcanic activity, and greenhouse gas concentrations (for more details see Bhend et al., 2012; Franke et al., 2017). The land-use reconstruction by Pongratz et al. (2008) was used to derive the land-surface parameters.

We would suggest only mentioning that the land-use reconstruction by Pongratz et al. (2008) was used in the paper because we would not like to give very different levels of detail for each forcing or boundary condition. The reconstruction by Pongratz is the standard land-use reconstruction used in basically all ECHAM model simulations since the reconstruction was published. For the Reviewer interest, the 14 land-cover classes defined by Pongratz et al (2008) were mapped to land surface parameters closely following the procedure of Hagemann et al. (1999).

**Page 4, line 16, 'CCC400' has not been introduced**

In the title of section 2.1 we try to indicate that the model simulation is called CCC400. It is not an abbreviation but the name of the model simulation. To make it more clear we will replace page 3, line 25-26 as: It uses a 30 member ensemble of atmospheric model simulations (CCC400) as background to reconstruct monthly climate states between 1600 and 2005.

**Page 4, line 23, If I understand correctly the deviations of the ensemble members from the ensemble mean are updated in the online EnSRF according to equation 2, but not in the offline EKF, which uses an existing ensemble. Please clarify.**

In both online and offline EnSRF methods, the mean and the deviation from the mean are updated, according to Eq. (1) and Eq. (2). However, in an online application the analysis and its uncertainty are propagated forward to the next time steps. In case of an offline approach the process stops after all available observations at the current time steps have been assimilated.

**Page 4, lines 28-30, the statements on the 6 month periods are partly redundant.**

We will rewrite it.

**Page 5, lines 5-7, How have the error variances been chosen? Should sigma^2 be K^2? If so, why is the error for documentary data smaller than for instrumental data? Which multiple regression?**

The error variances are rough estimates that include for instance measurement uncertainties, temporal inhomogeneities, and the fact the a station is not representative for a grid cell (see Franke et al. 2017 and Frei 2014).

The error variance of documentary indices without units is set to 0.25 standard deviations^2. Because in the experiments described in this paper, we did not assimilate documentary data, we will delete the information regarding to this source in the revised paper.

Will be replaced as:

We set the error variances of instrumental temperature observations to 0.9 K^2, and of instrumental pressure data to 10 hPa^2. The errors of tree-ring proxy data are calculated as the variance of the multiple regression residuals of the H operator.

With regard to "multiple regression" we write in section 2.2 that the H operator was designed as a multiple linear regression model to transform the model state to proxy records. We did not explain it in details only referred to the paper by Franke et al., 2017 (page 5. line 2-3). We will add further information about the H operator.

**Page 5, lines 16 – 19. The notation is not clean. In line 16 it is said that R is a diagonal matrix, in line 19 that R is a scalar. The problem is that the same notation is used for an equation using the full set of observations (where R is a diagonal matrix) and for the equation when the individual observations are assimilated sequentially. Please reformulate.**

It is true that the same notation is used for describing the whole system or the serial approach, but in other papers (including the original paper by Whitaker and Hamill, 2002) the notation remains unchanged. Therefore, we would like to follow the common usage of the notation and keep it as it was in the submitted paper.

**Page 6, line 1, replace 'localization function' with 'the localization function'.**

We will replace it.

**Page 6, line 19/20, replace 'additive inflation' with 'the additive inflation', and 'hybrid' with 'the hybrid'.**

We will replace it.

**Page 6, line 23-25, The explanation is confusing. One can select ensemble members for the whole period or some or all ensemble members for some time steps; how exactly are the climatological state vector and the associated error covariance matrix calculated? The simulations have already been performed; why are there substantial computational costs for using a large number of ensemble members?**

The climatological state vector is created as, 1) we define how many members $x^{clim}$ should have. We used 100, 250, and 500 members to create $x^{clim}$. 2) If the climatological state vector should have e.g. 250 members then 250 years are randomly selected from the whole model simulation, between 1601 and 2005. 3) Every year has 30 members from which one is randomly selected and kept. 4) The chosen members are combined in the climatological state vector.

The climatological background-error covariance matrix is calculated from this ensemble , by using the ensemble perturbations.

The climatological state vector is randomly resampled after every second assimilation cycle. The increased computational cost partly comes from the creation of the climatological state vector. The other time consuming part comes from the updating of the climatological part after each observation is assimilated. (The standard way when observations are assimilated serially).

**Page 6, line 29, $H^T$ is at the end of all terms in the equation. Can it not simply be deleted?**

Yes, it is possible to delete them.

**Page 7, line 4-5. It is not clear how $x^{clim}$ is calculated and updated, what n is, and what 'propagated' means in an offline assimilation scheme.**

We will give a better explanation how $x^{clim}$ is obtained and define n as the ensemble size used to create a climatological state vector.

We assimilate observations serially, that is observations are processed one at a time. After the first observation is assimilated and the analysis is obtained, this analysis field will become the background state for the next observation. On Figure 2 the arrows pointed from the analyses to the background state vectors are meant to indicate this process. Propagating and updating also refer to this process, that the information from the observation is also incorporated into $x^{clim}$ and not only in $x^b$ in most of our experiments (see Table 2).

We will reformulate the sentences to avoid confusion.

**Page 8, line 7, replace 'isotropic localization' with 'an isotropic localization'.**

We will replace it.

**Page 8, line 15, ENH is not defined.**

In page 8 line 2, we introduced ENH, which stands for extratropical Northern hemisphere.

**Page 9, line 3, Explain why P$^b$ does not have full rank, why this is a problem, and what this has to do with inflation.**

In the Introduction (page 2, line 20-29), some applications of covariance inflation were mentioned and how they help to avoid the negative effect of sampling error. Inflation techniques were discussed later again in section 3.2.

In page 9, line 3, we will emphasis once again the negative effect of the small ensemble size and referring back to section 3.2 how inflation techniques can help. We will reformulate the sentence using the term 'sampling error' because it has been introduced previously and it will make a better connection.

**Page 9, lines 5-6. This is not well phrased; it is not the covered model space that is multiplied with the inflation factor.**

Thank you for pointing out the mistake. We will reformulate the sentence by indicating that only the deviations (x') were multiplied.

**Page 9, line 16, It has not yet been mentioned in the main text in section 3 that 250 members have been chosen and how they have been selected (see also earlier comment on this).**

In most of our experiment, we indeed used 250 members to create the climatological state vector. We wrote that the ensemble size of the climatological part ranged between 100 and 500 members in the experiments We will clarify for the readers, how the climatological state vector is obtained.

**Page 12, line 2, replace 'verification' with 'validation' (this is used elsewhere in the text) or 'performance'**

We will replace it.

**Page 12, line 6, missing full stop after '2'.**

We will correct it.

**Page 12 and also in main text, Add a comment on whether the skill differences found are substantial and practically relevant.**

We agree that this statistic is missing and we will provide some statistical evaluation in the revised paper.

**Page 14, line 11, 'kalman' should be upper case.**

We will correct it.

**Page 14, line 16/17, typos and missing spaces**

We will correct it.

**Figs. 1, 3, 8, 10 should be bigger**

We will provide well readable figures in the final version.

References

Frei, C.: Interpolation of temperature in a mountainous region using nonlinear profiles and non-Euclidean distances, International Journal of Climatology, 34, 1585–1605, 2014.

---

## Author Response (AR1)

**Reviewer #1**

**This paper tests several different methodological choices that are typically made (or could be made) in paleoclimate reconstructions using DA. I think this presents a good and valuable presentation and discussion of these choices. The findings and suggestions for future reconstructions are very helpful to the community performing these types of reconstructions.**

We thank the Reviewer for the positive comments on our work and for the suggestion how to improve our figures.

**Section 3.1: Is there any specific justification for the choice of the ratio of L_z and L_m being 2:1? Could this, or some other ratio, be justified by looking at the correlation length scale in observational data?**

> The idea behind a longer length scale in zonal direction than in meridional direction is based on the zonal flow in the atmosphere. On multi-annual to multi-decadal time scales multiple processes act in meridional direction, e.g. a widening/shrinking of the Hadley cell, shifts of the ITCZ or changes in atmospheric modes like the AMO or the NAO. These can shift the zonal circulation northward or southward but the zonal coherence will be less affected.

> In principle the correlation length scale found in observational data or reanalyses could be used, but the anisotropy depends strongly on season, variable and location (e.g., it is much larger in the tropics than in the extratropics). Between experiments "no localization" (of Pclim) and localization of the standard setup, a 2:1 experiment seems like a good intermediate.

> We explained the hypothesis in more details in the revised manuscript.

**Section 3.1: Is there any justification for the specific localization values that you chose for each variable that was reconstructed? Are these values data-driven or just educated guesses? Were any experiments done to test on optimal localization value? I would assume that if these values were used based on weather DA experiments, they might not apply on the longer paleo time scales where one would generally expect the correlation length scales to be larger.**

> The localization length scale parameters were defined based on the spatial correlation of the variables in the monthly ECHAM model simulation fields. In Section 2.4 we refer to the paper by Franke et al. 2017 how the localization was done in the original setup. We used the same localization length scale parameters for localizing the sample covariance in most of our experiments to evaluate improvements in comparison with this initial setup. For this study, we calculated the latitudinal dependency of correlation of the state variables from a bigger ensemble of the model than in Franke et al (2017). The result suggested that the longer length scale parameters can be applied in the tropics and the predefined length scale parameter of precipitation is probably too strict. Based on the rather strict correlation length scale parameters in the previous study and the assumption that the covariances can be better estimated from a bigger ensemble, we used doubled length scale parameters in some of the experiments for localizing the climatological covariances. In this case, the L for temperature is 3000 km, which means that the correlation is decreased close to zero approximately 6000 km away from the observation. We did not carry out further experiment to find the optimal localization value because even double the localization distance hardly changed the reconstruction skill, not even at locations that newly fall inside the radius of influence of at least a single observation. Hence, our setup does not appear to be very sensitive on the localization distance as long as it remains in a reasonable range. On the one hand, we do not

further restrict the localization because that would limit updates to a small regions around observations. On the other hand, our experiments without localization showed negative reconstruction skill in locations far away from observations, even with the error covariance matrix is calculated from climatology. We provide further explanation about localization in Section 2.4 and Section 3.2 in the revised manuscript.

**Section 4.2.1: When you are comparing the distributions, you say that for example,the most skillful reconstruction is obtained from the 100c_PcL experiment. What is the basis for saying it's the best? What aspect of the distribution are you comparing? The median or some other specific value(s)?**

Yes, for comparison we used only the median.

**Many of the distributions shown in the figures look very similar so it was hard for me to feel confident about the statement that one particular set of reconstruction choices was better than another. Are the distributions statistically distinct?**

We agree with the Reviewer that the distributions of the skill of the experiments over the extratropical Northern hemisphere look similar. We have not checked whether the distributions are statistically distinct. In the revised paper we provide an evaluation of the skill scores of the experiments compared to the original analysis skills, using a permutation test.

**Instead of comparing the distributions, would it be possible to show the differences compared to the "original"reconstruction (i.e., you'd compute the difference in the skill score for each location and then summarize this distribution of differences in the plots)? I'm wondering if this,or something similar, might make the differences more clear. Because currently when I look at the distributions, many of them look very similar and perhaps even statistically indistinguishable.**

Thank you for your suggestion. We made the plots as it was suggested and replaced Fig. 4-6 and Fig. 8 showing the differences between the experiments and the original analysis.

**Fig 8 & 10: It would be very helpful to give a little more explanatory information/labeling on each panel, such as was done in Fig 3.**

We added more labels to the figures (in the revised manuscript Fig. 7 and Fig. 9).

**Reviewer #2**

**This manuscript identifies the best choices from a number of different spatial and temporal localization approaches and from different inflation techniques for the background error covariance matrix in Ensemble Kalman Filters used in paleoclimatic applications. The optimization of these technical details in data assimilation is important for the growing paleoclimate data assimilation community. The results are systematically derived and the manuscript is in general well written. I support publication after the points listed below have been clarified or corrected.**

We thank the Reviewer for the careful revision of the manuscript. We will follow his/her recommendations and add further details at the indicated points to provide better understanding how ensemble Kalman filter methods work, how we set up the blending experiment and how covariance inflation techniques can help to better estimate the background-error covariance matrix. We are also thankful for improving the English of the manuscript.

**Specific comments**

**Page 1, Line 2, replace 'of the assimilation system' with 'of some assimilation systems'**

> We replaced it.

**Page 1, Lines 17/18, 24/25, 'boundary conditions' are specifications of state variables at the boundaries of a model domain, and thus not the same as 'forcings', which are external influences on the system. The two should be distinguished throughout the text. It seems that here the statement are about forcings. If so, reformulate avoiding the use of 'boundary condition'.**

> Was replaced as:

> Climate models constrained with realistic, time-dependent external forcings provide fields that are consistent with these forcings and the model physics.

> If information from observations is optimally blended with climate model simulations, the result is the best estimate of the climatic state, given the observations, given the external forcings, and given the known climate physics.

**Page 2, line 6, 'linear models' of what? I think it should by 'linear dynamical systems'. A short comment on why KFs are used with non-linear systems, including GCMs, would be good. 'Gaussian distributions' of what? The state variables?**

> Yes, the KF is optimal for linear systems (linear dynamical model and linear observation operator) with Gaussian error distributions (model, background, and observation). In atmospheric data assimilation the most commonly used KFs are the ensemble-based Kalman filters that can handle some non-linearity in the dynamics and in the observation operator. Moreover, in our approach we use an offline implementation of the ensemble square root filter, i.e. we only adjust the precomputed model simulations with the observations and never deal with the question of model dynamics. Nevertheless, we removed this sentence because it was not necessary here and just caused confusion.

**Page 2, line 13-16, I suggest using 'stationary offline' and 'transient offline' for the two approaches.**

Thank you for the suggestion. We used the terms as stationary (forcing-independent) offline and transient (forcing-dependent) offline to better distinguish between the two methods.

**Page 2, lines 17/18, 'The true climate state is not known, therefore it has to be estimated'. Does 'it' refer to 'the true climate state', as the sentence suggests or to 'the uncertainty of the background state', which would link better to the first sentence in this paragraph? This sentence should be clarified, or it could simply be deleted (which I think is the better option).**

We deleted the sentence.

**Page 2, line 20, What is a non-simplified KF? A KF with a 'true' background error covariance matrix? If so, how can this exist? The background error covariance has always to be estimated somehow. Please clarify the statement.**

We repalced 'Ensemble-based KFs are simplification of the KF' to 'Ensemble-based KFs are approximations of the KF'.

In the Kalman filter, the background-error covariance matrix is given as

$$P^b = \overline{(x^b - x^t)(x^b - x^t)^T}$$

the superscripts b and t indicate the background state and the true state, while the overline marks the expectation value.

In practice, as was said, the background-error covariance matrix has to be estimated. In ensemble-based Kalman filter techniques, the ensemble covariance will provide the background-error covariance matrix and the true state is estimated by the ensemble mean.

$$P^b_e = \overline{(x^b - \overline{x^b})(x^b - \overline{x^b})^T}$$

**Page 2, line 23, It seems that the sampling error for the background error is not only a random error, but leads to a systematic underestimation of the background error, otherwise inflation would not be a suitable approach. Please explain better.**

Ensemble-based filters with flow-dependent covariances can diverge due to sampling error. Our small 30 member ensemble may result in an error underestimation, because we do not assess the uncertainties in SST boundary conditions and external forcings. In an online ensemble-based KF approach, after the update step all ensemble members are propagated forward according to the model dynamics. However, if the uncertainty of the analysis is underestimated in the update step, the background error may be underestimated in the next time step and the method will trust the model more and more, while giving less and less weight to the observations in the following time steps. This should not be a problem in offline approaches. However, we conduct this experiment for completeness because as we say in the manuscript, there are two commonly used methods to reduce the negative effect of sampling error: inflation (e.g., Anderson and Anderson, 1999), and localization (e.g., Hamill et al., 2001) of the background-error covariance matrix."

The multiplicative inflation part was rewritten as:

A simple inflation technique is the multiplicative inflation (Anderson and Anderson, 1999), which compensates for potential underestimation of the analysis error. Multiplicative

inflation helps to maintain a more realistic distribution of the ensemble members by increasing the deviation of the members from the ensemble mean at each DA cycle (Anderson and Anderson, 1999). However, the underestimation of the analysis error is of minor importance in offline approaches, because the ensemble members are not propagated forward in time.

**Page 2, lines 25/26. The statement on distribution of ensemble members refers to online approaches, but the approach used by the authors is an offline approach. This is confusing. Please briefly explain how the ensembles are generated in an online KF, and that in offline approaches the ensemble is given, but that the background error covariance still needs to be inflated.**

To avoid any confusion, we changed the sentence to: "Two methods are commonly used in online ensemble-based KF approaches to reduce the negative effect of sampling error:" On page 2, line 8-11 the main steps of an online KF is already described. These steps remain the same in online ensemble-based Kfs. In the update step the ensemble members are updated with the observation when they become available. In the forecast step these updated ensemble members are propagated forward by the model to the next time step, then the mean and the covariance are estimated again. Finally, with regard to the need to inflate the background error covariance, please see our answer to the last comment on page 2, line 23.

**Page 3, line 14, replace 'other method' with 'additive method'**

We replaced it.

**Page 3, line 25, replace ''form' with 'from'**

We replaced it.

**Page 3, line 28, Don't use 'forced by boundary conditions', as forcings and boundary conditions are different (see comment above). If I understand correctly for all ensemble members the same greenhouse gas, solar and volcanic forcings have been used, as well as the same SST boundary conditions. Please clarify.**

Yes, the same forcings and boundary conditions were used in the 30 members.

Was replaced as:

The 30 ensemble members were forced and driven with the same external forcings and with the same boundary conditions.

**Page 3, Lines 29-31, The SST reconstruction can be expected to strongly influence the results of this data assimilation approach with an atmosphere-only GCM. There should be some comments on how the SST reconstructions have been made, what is known about their uncertainties, and why this approach is taken rather than data assimilation with a coupled atmosphere-ocean GCM.**

We use the SSTs from Mann et al. (2009), augmented as described in Bhend et al. (2012). The main reason why we use the SST reconstruction by Mann et al. 2009 is that it was the only available reconstruction at the time when we ran the simulation ensemble, like we already explained in the paper: "For sea-surface temperatures (SSTs), which have a

particularly large effect on the simulations, the reconstruction by Mann et al. (2009) was used. This is the only global gridded SST reconstruction that dates back till 1600."

We already describe limitations of the reconstruction and how we tried to improve them in the discussion paper, too: "The SST reconstruction by design captures interdecadal variations (Mann et al., 2009), hence intra-annual variability dependent on a El Niño/Southern Oscillation reconstructions (Cook et al.,2008) was added to the SST fields." We will add a reference to the paper of Bhend et al. 2012, which explains the simulations in a little more detail.

There are multiple reasons why we do not use a coupled atmosphere-ocean model:

1. We understand a paleo-reanalysis to be a product that best describes the atmospheric states of the past globally. 400 years of just externally forced coupled atmosphere-ocean simulations would not be in close agreement with the true state of the ocean in the past and e.g. not produce El Niño events in the years when they actually occurred. Therefore such simulations would not capture related teleconnections patterns in the corresponding years, either. We can not gain the same information that currently comes from the SST boundary conditions with our data assimilation setup because we only assimilate absolutely dated tree-ring proxies on land and mostly at mid to high latitude. It would be an option to assimilate coral data in the future but there are few records that back to 1600 and their assimilation has not been tested yet. That is why coral data assimilation in not an option for the presented sensitivity experiments. The focus of this study is to show, how the paleo data assimilation setup published by Franke et al. (2017) can be further improved without introducing further changes that would blur the effect of the improved covariance estimation.

2. If we would like the ocean to be in closer agreement with the observations in a coupled model, we would need to do an online assimilation scheme that could take the longer memory of the ocean into account whereas our atmospheric states have no memory that extends beyond a month or year when the next proxy observation becomes available. Hence, running a coupled ocean-atmosphere model would not allow us to work with our offline assimilation system and do all the sensitivity experiments, which we present in this study.

3. Finally, an ensemble of coupled simulations plus the additional spin-up time for the ocean to reach equilibrium would have simply not been computationally affordable.

**Page 3, line 30, I think 'till' should not be used in formal writing and should be replaced with 'until' (also later in the text).**

We replaced it, also in page 4, line 10-12

**Page 4, line 1, replace 'boundary conditions' with 'forcing', and make a separate statement on land-surface boundary conditions, including which variables are prescribed.**

Was replaced as:

Further forcings include solar irradiance, volcanic activity, and greenhouse gas concentrations (for more details see Bhend et al., 2012; Franke et al., 2017). The land-use reconstruction by Pongratz et al. (2008) was used to derive the land-surface parameters.

We would suggest only mentioning that the land-use reconstruction by Pongratz et al. (2008) was used in the paper because we would not like to give very different levels of detail for each forcing or boundary condition. The reconstruction by Pongratz is the standard land-use

reconstruction used in basically all ECHAM model simulations since the reconstruction was published. For the Reviewer interest, the 14 land-cover classes defined by Pongratz et al (2008) were mapped to land surface parameters closely following the procedure of Hagemann et al. (1999).

**Page 4, line 16, 'CCC400' has not been introduced**

In the title of section 2.1 we try to indicate that the model simulation is called CCC400. CCC400 stands for "Chemical climate change over the past 400 years". The name is misleading since the model was run without chemistry due to a lack of computing resources.

To make it more clear we will replace page 3, line 25-26 as:
The model simulation, termed as Chemical Climate Change over the Past 400 years (CCC400), has 30 ensemble members, that are used as background to reconstruct monthly climate states between 1600 and 2005.

**Page 4, line 23, If I understand correctly the deviations of the ensemble members from the ensemble mean are updated in the online EnSRF according to equation 2, but not in the offline EKF, which uses an existing ensemble. Please clarify.**

In both online and offline EnSRF methods, the mean and the deviation from the mean are updated, according to Eq. (1) and Eq. (2). However, in an online application each ensemble member is propagated forward to the next time steps. In case of an offline approach the process stops after all available observations at the current time steps have been assimilated.

**Page 4, lines 28-30, the statements on the 6 month periods are partly redundant.**

Was replaced as:

In the EKF, the length of the assimilation window is 6 month (October-March and April-September), which were adapted to the southern and northern hemispheric growing seasons to effectively incorporate the proxy records stored in trees. Due to the 6-monthly assimilation window, $x^b$ contains the variables of 6 months.

**Page 5, lines 5-7, How have the error variances been chosen? Should sigma^2 be K^2? If so, why is the error for documentary data smaller than for instrumental data? Which multiple regression?**

The error variance of documentary indices without units is set to 0.25 standard deviations^2. Because in the experiments described in this paper, we did not assimilate documentary data, we will delete the information regarding to this source in the revised manuscript.

Was replaced as:

We set the error variances of instrumental temperature observations to 0.9 K^2, and of instrumental pressure data to 10 hPa^2. The error variances are rough estimates that include for instance measurement uncertainties, temporal inhomogeneities, and the fact that a station is not representative for a grid cell (see Frei, 2014; Franke et al., 2017). The errors of tree-ring proxy data are calculated as the variance of the multiple regression residuals of the H operator.

We added further explanation about the H operator:

H is the forward operator that maps the model state to the observation space (here, it is linear). H differs depending on the type of observation being assimilated. In case of tree-ring width data H extracts temperature between May and August and precipitation between April and June from the model, then these fields are transformed to observational space by using a multiple regression approach (for more details see Franke et al., 2017).

**Page 5, lines 16 – 19. The notation is not clean. In line 16 it is said that R is a diagonal matrix, in line 19 that R is a scalar. The problem is that the same notation is used for an equation using the full set of observations (where R is a diagonal matrix) and for the equation when the individual observations are assimilated sequentially. Please reformulate.**

It is true that the same notation is used for describing the whole system or the serial approach, but in other papers (including the original paper by Whitaker and Hamill, 2002) the notation remains unchanged. Therefore, we would like to follow the common usage of the notation and keep it as it was in the submitted paper.

**Page 6, line 1, replace 'localization function' with 'the localization function'.**

We replaced it.

**Page 6, line 19/20, replace 'additive inflation' with 'the additive inflation', and 'hybrid' with 'the hybrid'.**

We replaced it.

**Page 6, line 23-25, The explanation is confusing. One can select ensemble members for the whole period or some or all ensemble members for some time steps; how exactly are the climatological state vector and the associated error covariance matrix calculated? The simulations have already been performed; why are there substantial computational costs for using a large number of ensemble members?**

We added the following explanation:

The climatological state vector is created as: 1. Define the ensemble size (n) of $x^{clim}$ ; 2. Select n random years between 1601 and 2005; 3. Every year has 30 members from which one member is randomly selected and kept; 4. The chosen members are combined in $x^{clim}$ . $x^{clim}$ is randomly resampled after every second assimilation cycle. Using $x^{clim}$ in the assimilation leads to increased computational cost, which partly comes from the creation of $x^{clim}$. The other time consuming part comes from the updating of the climatological part after each observation is assimilated. (The standard way when observations are assimilated serially). We tested numbers between 100 and 500. From $x^{clim}$ a climatological background-error covariance matrix ($P^{clim}$) can be obtained by using the ensemble perturbations.

**Page 6, line 29, $H^T$ is at the end of all terms in the equation. Can it not simply be deleted?**

We deleted them.

**Page 7, line 4-5. It is not clear how $x^{clim}$ is calculated and updated, what n is, and what 'propagated' means in an offline assimilation scheme.**

Please see our reply to comment Page 6, line 23-25.

We assimilate observations serially, that is observations are processed one at a time. After the first observation is assimilated and the analysis is obtained, this analysis field will become the background state for the next observation. On Figure 2 the arrows pointed from the analyses to the background state vectors are meant to indicate this process. Here, by propagating we refer to this process, that the information from the observation is also incorporated into $x^{clim}$ and not only in $x^b$ in most of our experiments (see Table 2).

We replaced propagate with update to avoid confusion.

**Page 8, line 7, replace 'isotropic localization' with 'an isotropic localization'.**

We replaced it.

**Page 8, line 15, ENH is not defined.**

In page 8 line 2, we introduced ENH, which stands for extratropical Northern hemisphere.

**Page 9, line 3, Explain why $P^b$ does not have full rank, why this is a problem, and what this has to do with inflation.**

In the Introduction (page 2, line 20-29), some applications of covariance inflation were mentioned and how they help to avoid the negative effect of sampling error. Inflation techniques were discussed later again in section 3.2.

We rewrote the sentence as:

The main problem of ensemble-based DA techniques is the computationally affordable limited ensemble size. Due to the finite ensemble size the estimation of $P^b$ suffers from sampling error. Applying inflation techniques is one method to mitigate its effect (see Sec. 3.2).

**Page 9, lines 5-6. This is not well phrased; it is not the covered model space that is multiplied with the inflation factor.**

Thank you for pointing out the mistake.

We rewrote the sentence as:

Using the multiplicative inflation method, the deviations from the ensemble mean are multiplied with a small factor (γ).

**Page 9, line 16, It has not yet been mentioned in the main text in section 3 that 250 members have been chosen and how they have been selected (see also earlier comment on this).**

In most of our experiment, we indeed used 250 members to create the climatological state vector. We wrote that the ensemble size of the climatological part ranged between 100 and 500 members in the experiments.

Here, we added a sentence:

In most of our experiment n is 250.

**Page 12, line 2, replace 'verification' with 'validation' (this is used elsewhere in the text) or 'performance'**

We replaced it.

**Page 12, line 6, missing full stop after '2'.**

We corrected it.

**Page 12 and also in main text, Add a comment on whether the skill differences found are substantial and practically relevant.**

We agree that this statistic was missing. In the revised paper we provide an evaluation of the skill scores of the experiments compared to the original analysis skills, using a permutation test.

**Page 14, line 11, 'kalman' should be upper case.**

We corrected it.

**Page 14, line 16/17, typos and missing spaces**

We corrected it.

**Figs. 1, 3, 8, 10 should be bigger**

We replaced the figures in the revised version.

References

Frei, C.: Interpolation of temperature in a mountainous region using nonlinear profiles and non-Euclidean distances, International Journal of Climatology, 34, 1585–1605, 2014.

Relevant changes

- All figures were modified and Figure 4 and Figure 5 were combined into one Figure.

-  Eq. 1, Eq. 2 and Eq. 6 were rewritten.

- A permutation test was conducted to test significant difference between the experiments and the original version.

[revised manuscript text omitted]